# Dbi1 is an oxidoreductase and an assembly chaperone for mitochondrial inner membrane proteins

Soraya Badrie[1], Kai Hell[2] & Dejana Mokranjac [ID] [1✉]

## Abstract

Import and assembly of mitochondrial proteins into multimeric complexes are essential for cellular function. Yet, many steps of these processes and the proteins involved remain unknown. Here, we identify a novel pathway for disulfide bond formation and assembly of mitochondrial inner membrane (IM) proteins. Dbi1, a previously uncharacterized IM protein, interacts with an unassembled pool of Tim17, the central subunit of the presequence translocase of the IM, and is upregulated in cells with increased levels of unassembled Tim17. In the absence of Dbi1, the conformation of the presequence translocase is affected and stability of Tim17 is reduced. Furthermore, Dbi1, through its conserved CxxC motif, is involved in the formation of the disulfide bond in Tim17 in a manner independent of the disulfide relay system, the major oxidation-driven protein import pathway into mitochondria. The substrate spectrum of Dbi1 is not limited to Tim17 but includes at least two more IM proteins, Tim22 and Cox20. We conclude that Dbi1 is a novel oxidoreductase in mitochondria which introduces disulfide bonds into IM proteins and chaperones their assembly into multimeric protein complexes.

Keywords Assembly Chaperone; Disulfide Bond; Mitochondria; TIM22 Complex; TIM23 Complex
Subject Categories Membranes & Trafficking; Organelles

## Introduction

The vast majority of cellular proteins need to be translocated across at least one cellular membrane and subsequently assembled into multimeric protein complexes before they can fulfill their biological functions (Alberts, 1998; Michaelis et al, 2023; Wickner and Schekman, 2005). The function of many proteins requires further modifications such as for example introduction of disulfide bonds. Yet, despite many years of intensive research, many steps in these processes and the proteins involved still remain unknown.

With the exception of a handful of proteins encoded in the mitochondrial genome, about 1000 different mitochondrial proteins are synthesized in the cytosol as precursor proteins and subsequently imported into the organelle. Mitochondria contain an elaborate network of protein translocases that recognize and sort mitochondrial proteins into one of the four mitochondrial subcompartments, the outer membrane, the inner membrane (IM), the intermembrane space (IMS), and the matrix (Araiso et al, 2022; Busch et al, 2023; Drwesh and Rapaport, 2020; Hansen and Herrmann, 2019). The majority of mitochondrial precursor proteins carry N-terminal presequences (Vogtle et al, 2009) and are translocated into the matrix or inserted into the IM with the help of the TIM23 complex, also known as the presequence translocase. Many hydrophobic inner membrane proteins, such as members of the metabolite carrier family and central components of the IM protein translocases, have no presequences and are inserted into the IM with the help of the TIM22 complex, also known as the carrier translocase. Small, soluble IMS proteins with conserved cysteine motifs, typically twin $Cx_3C$ and twin $Cx_9C$, follow the oxidation-driven import pathway mediated by the disulfide relay system consisting of the oxidoreductase Mia40 and the sulfhydryl oxidase Erv1 (Edwards et al, 2020; Hell, 2008; Riemer et al, 2009; Stojanovski et al, 2012). Mia40 is both a receptor and an oxidoreductase. By forming a transient intermolecular disulfide bond with the incoming polypeptide chain through its redox active CPC motif, Mia40 oxidizes its substrates. This newly introduced disulfide bond facilitates folding of the newly imported protein and thus its retention in the IMS. Erv1 then re-oxidizes Mia40 for another round of import. Still, a number of mitochondrial proteins contain disulfide bonds that are not introduced by the disulfide relay system (Riemer et al, 2009). How they are oxidized remains unresolved.

Both TIM23 and TIM22 complexes are intricate molecular machines consisting of a number of different subunits. Whereas a number of proteins involved in the assembly of respiratory chain complexes and mitochondrial ribosomes were characterized in the past (Fernandez-Vizarra and Zeviani, 2021; Khawaja et al, 2023; Vercellino and Sazanov, 2022), no protein involved in the assembly of the two major protein translocases of the inner membrane was identified to date and how these essential molecular machines are assembled is completely unknown. The currently known subunits

[1]LMU Munich, Biozentrum—Cell Biology, 82152 Planegg-Martinsried, Germany. [2]LMU Munich, Biomedical Center—Physiological Chemistry, 82152 Planegg-Martinsried, Germany. ✉E-mail: mokranjac@bio.lmu.de

of the TIM23 complex are Tim50, Tim23, Tim17, Tim44, mtHsp70, Tim14(Pam18), Tim16(Pam16), Mge1, Mgr2, Tim21, and Pam17 (Genge and Mokranjac, 2021; Schulz et al, 2015). They are operationally divided into the IMS-exposed receptors which recognize presequences in the IMS (Tim50 and Tim23) (Caumont-Sarcos et al, 2020; Geissler et al, 2002; Genge et al, 2023; Schulz et al, 2011; Yamamoto et al, 2002), the membrane-embedded core of the complex which forms the translocation path across the IM (Tim17 and Tim23) (Fielden et al, 2023; Sim et al, 2023) and the import motor on the matrix-side of the IM which completes ATP-dependent translocation into the matrix (Tim44, mtHsp70, Tim14, Tim16, and Mge1) (Craig, 2018; Mokranjac, 2020). Mgr2, Tim21, and Pam17 modulate the function of the TIM23 complex and its association with the respiratory chain complexes. The central components of the TIM22 complex are membrane-embedded Tim22 and the hexameric small TIM chaperones associated with Tim22 on the IMS side (Busch et al, 2023; Qi et al, 2021; Zhang et al, 2021). Further subunits of the TIM22 complex are involved in the recruitment of the small TIM chaperones to Tim22 and in the stability of the complex and they differ between fungi and metazoa. Tim17, Tim23, and Tim22 belong to the same protein family, and each contains four transmembrane (TM) domains (Zarsky and Dolezal, 2016). The conformations of Tim17 and Tim22 are stabilized through evolutionarily conserved structural disulfide bonds formed between two cysteine residues on the IMS sides of the proteins (Okamoto et al, 2014; Ramesh et al, 2016; Wrobel et al, 2016; Wrobel et al, 2013). In both cases, the disulfide bonds are not essential for cell viability but play important roles in the stability of the complexes and contribute to the activity of the respective translocases, especially under stress conditions (Okamoto et al, 2014; Ramesh et al, 2016). Mia40 is involved in import of both Tim17 and Tim22, however, this oxidoreductase does not appear to have a role in formation of the disulfide bond in either Tim17 or Tim22 (Okamoto et al, 2014; Ramesh et al, 2016; Wrobel et al, 2016; Wrobel et al, 2013). How these central components of the two major mitochondrial protein translocases get their disulfide bonds is unknown.

By looking into a previously uncharacterized crosslink within the TIM23 complex, we identify here Dbi1 (for disulfide bond formation in inner membrane proteins). Our results suggest that Dbi1 is a novel oxidoreductase in mitochondria involved in disulfide bond formation in Tim17 and in chaperoning its assembly into the TIM23 complex. Tim22 and Cox20 are further substrates of this newly identified pathway.

## Results and discussion

### Dbi1 is a novel crosslinking partner of the TIM23 complex

Crosslinking of Tim23 in intact mitochondria is a sensitive sensor of the conformational changes within the TIM23 complex (Banerjee et al, 2015; Popov-Celeketic et al, 2008). We recently observed (Gunsel et al, 2020) an unknown crosslink of Tim23 of ca. 37 kDa in the *tim23-87A5* mutant (Fig. 1A). We reasoned that this new crosslinking partner should be a conserved but poorly characterized mitochondrial protein of ca. 14 kDa. A search of the publicly available yeast databases revealed that the protein encoded by ORF

*YDL157c* fulfilled these requirements. Furthermore, its genetic profile resembled the ones of many TOM and TIM23 components (Usaj et al, 2017). To check whether the encoded protein indeed represents the crosslinking partner of Tim23 in *tim23-87A5* mutant mitochondria, we genomically tagged the protein with a C-terminal His-tag in both wild type (WT) and *tim23-87A5* background. When YDL157c was tagged in the *tim23-87A5* background, the uncharacterized crosslink was present but migrated slower, consistent with the size shift introduced by the His-tag (Fig. 1B). More importantly, when crosslinked mitochondria were solubilized in an SDS-containing buffer to dissociate all noncovalent interactions and incubated with NiNTA-agarose beads, only the ca. 37 kDa crosslink was specifically retained on the beads (Fig. 1B), unambiguously confirming that the protein encoded by *YDL157c* is the protein we searched for. We named the encoded protein Dbi1, for disulfide bond formation in inner membrane proteins, for the reasons explained later. Various bioinformatic tools predict Dbi1 to be a conserved, 118 amino acid residues long protein lacking a presequence but containing two TM domains present centrally within the sequence (Figs. 1C and EV1A,B). A submitochondrial localization experiment, using an antibody generated against the C-terminal peptide of Dbi1, showed that Dbi1 is accessible to an externally added protease only upon opening of the outer membrane and remains in the pellet after carbonate extraction (Fig. 1D). We made essentially the same observation when we analyzed submitochondrial localization of $^{35}$S-labeled Dbi1 that was imported into isolated mitochondria in vitro (Fig. 1E). These results demonstrate that Dbi1 is an integral IM protein which exposes both its termini into the IMS (Fig. 1F). Deletion of Dbi1 did not affect the levels of any mitochondrial protein analyzed (Fig. 1G). However, the crosslinking pattern of Tim23 was changed in mitochondria isolated from cells lacking Dbi1, compared to WT (Fig. 1H), demonstrating that Dbi1 influences the conformation of the TIM23 complex.

### Dbi1 interacts with the unassembled pool of Tim17

To obtain insight into association of Dbi1 with the TIM23 complex, we performed a coimmunoprecipitation experiment. Wild type mitochondria were solubilized with digitonin-containing buffer and incubated with the affinity-purified antibodies to the various TIM23 subunits prebound to Protein A-Sepharose beads. We used antibodies to Tim50 as the receptor of the TIM23 complex, to Tim17 and Tim23 as the core subunits of the complex and to Tim14 as the subunit of the import motor. Antibodies from a preimmune serum were used as a negative control. The precipitation pattern of Dbi1 differed from all other subunits of the TIM23 complex—it was exclusively precipitated with antibodies to Tim17 (Fig. 2A). All other subunits of the TIM23 complex were precipitated in equal amounts with both Tim17 and Tim23 antibodies, as described previously (Banerjee et al, 2015; Mokranjac et al, 2003b). This unexpected precipitation pattern of Dbi1 was confirmed in the reciprocal experiment—antibodies to Dbi1 depleted Dbi1 from the mitochondrial lysate and coimmunoprecipitated only Tim17 among all TIM23 subunits analyzed (Fig. 2B). These results revealed a small, previously unseen pool of Tim17 that is not assembled with Tim23, suggesting that Dbi1 may be an assembly chaperone of Tim17. If this were the case, we reasoned that the fraction of Tim17 bound to Dbi1 would be larger in a

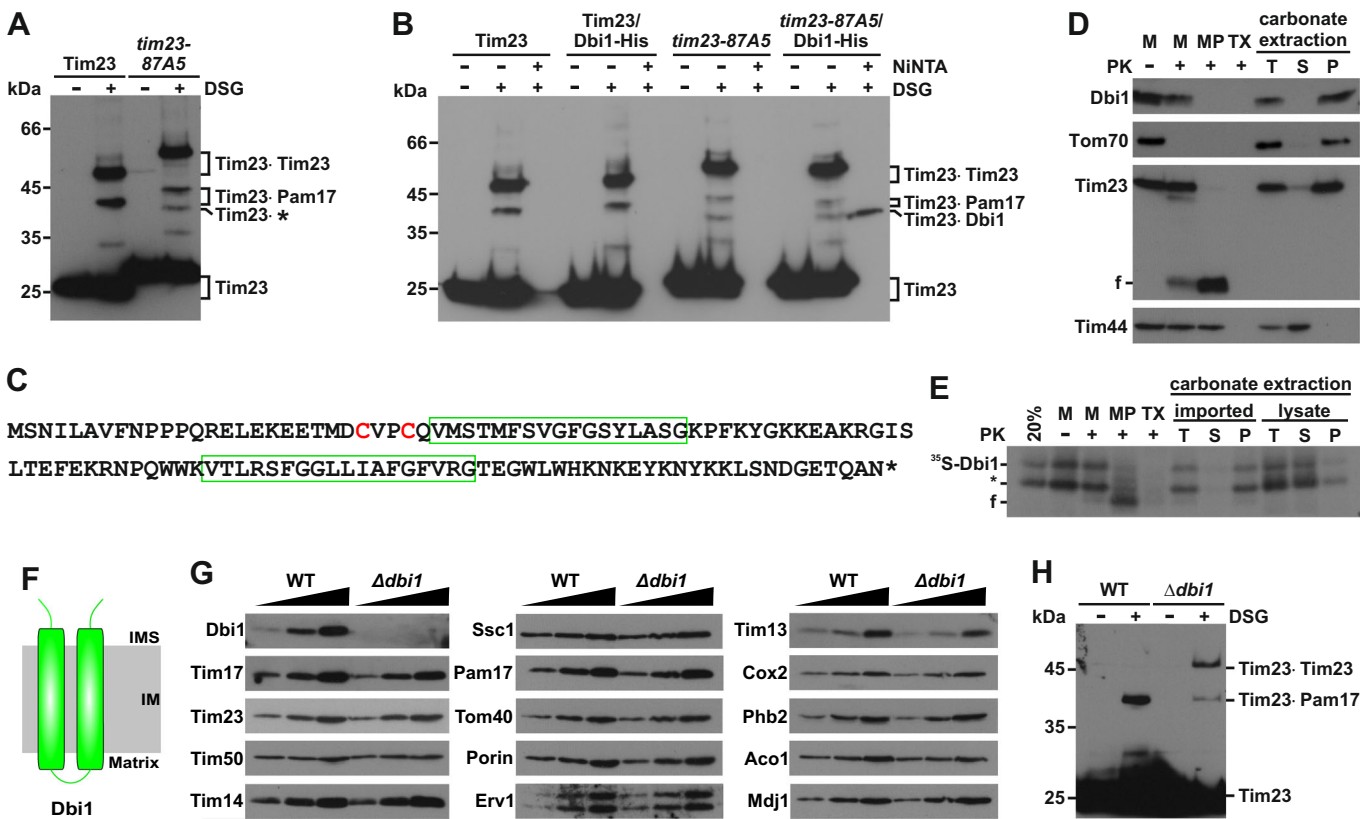

**Figure 1. Dbi1 is a novel interactor of the TIM23 complex.**

(A) Isolated mitochondria were crosslinked with DSG and analyzed by western blotting. * indicates the uncharacterized crosslink of Tim23. (B) Isolated mitochondria were crosslinked with DSG. Two samples were directly analyzed by western blotting. The third sample was solubilized in SDS-containing buffer and incubated with NiNTA-agarose beads. The bound fraction was analyzed by western blotting. (C) Protein sequence of Dbi1. Two predicted transmembrane segments and the conserved CxxC motif are highlighted. (D) Isolated wild type mitochondria (M), mitoplasts (MP) prepared by osmotic shock, Triton-solubilized mitochondria (TX) and, total (T), supernatant (S) and pellet (P) fractions of carbonate extraction were incubated with or without proteinase K (PK). Samples were analyzed by western blotting with the indicated antibodies. Tom70, Tim23, and Tim44 were used as markers for the outer membrane, the inner membrane and the matrix, respectively. f, a stable fragment of Tim23 in the inner membrane detectable with Tim23C antibody. (E) $^{35}$S-labeled Dbi1 was imported into isolated wild type mitochondria for 20 min and the samples were subsequently treated as in (D). Carbonate extraction was additionally performed with the translated lysate before import. Samples were analyzed by SDS-PAGE followed by autoradiography. * indicates the translation product starting from an internal methionine. f, a stable fragment of Dbi1 in the inner membrane. (F) Schematic representation of Dbi1 topology. (G) 10, 20, and 50 μg of isolated mitochondria (protein amount) were analyzed by western blotting with the indicated antibodies. (H) Indicated mitochondria were treated and analyzed as in (A). Source data are available online for this figure.

Tim23 mutant in which the Tim17–Tim23 interaction is destabilized, like, for example, *tim23G112L* (Demishtein-Zohary et al, 2015). Indeed, coimmunoprecipitation experiments demonstrated that both the fraction of Dbi1 precipitated with antibodies to Tim17 and the fraction of Tim17 precipitated with antibodies to Dbi1 were increased in the mutant mitochondria, as compared to the control (Fig. 2C). To shift the equilibrium towards free Tim17 even further, we used mitochondria isolated from cells in which expression of Tim23 was shut off for 21 h prior to mitochondria isolation (Popov-Celeketic et al, 2008). Whereas in WT mitochondria considerably larger amounts of Dbi1 and Tim17 were precipitated with their respective antibodies, the two antibodies precipitated essentially the same amounts of Dbi1 and Tim17 in mitochondria lacking Tim23 (Fig. 2D). Inspection of the proteins remaining in the nonbound fractions after coimmunoprecipitation revealed that in mitochondria lacking Tim23, antibodies to Dbi1 essentially immunodepleted Tim17 and vice versa (Fig. 2E). Interestingly, in mitochondria lacking Tim23, levels of Dbi1 were increased,

compared to the WT control (Fig. 2F), possibly to stabilize the free pool of Tim17. This resembles the situation when cells upregulate their chaperone network under stress conditions to deal with the increased number of unfolded proteins (Hartl et al, 2011; Rosenzweig et al, 2019).

Taken together, these results suggest that Dbi1 is not a genuine component of the TIM23 complex but rather the first identified assembly chaperone of the complex. Interestingly, the human homolog, DMAC1, was found associated with the assembly intermediates of complex I (Stroud et al, 2016).

## Dbi1 introduces the disulfide bond into Tim17

We found the specific association of Dbi1 with Tim17 and not with Tim23 rather surprising since Tim17 and Tim23 are structurally related proteins. There are, however, two noteworthy differences between Tim23 and Tim17. Tim23 contains a large, intrinsically disordered domain in the IMS that is absent in Tim17 (Fig. 3A). On

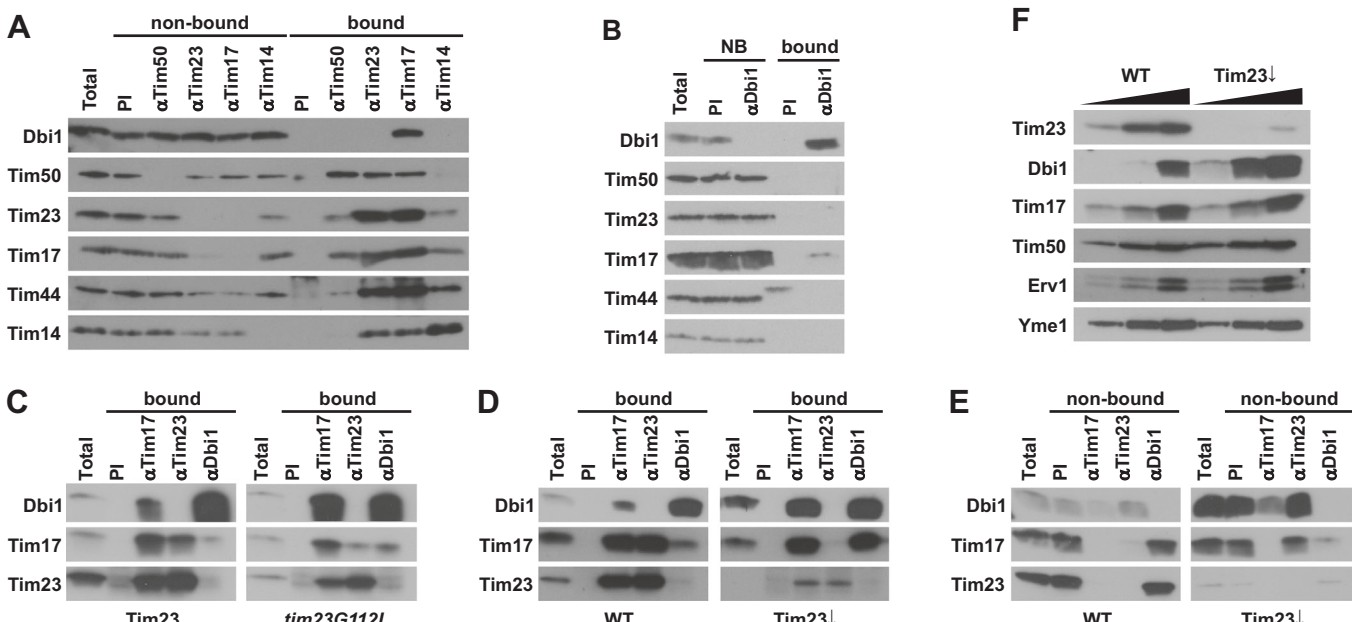

Figure 2. Dbi1 interacts with the unassembled pool of Tim17.

(A) Isolated wild type mitochondria were solubilized in digitonin-containing buffer and subjected to coimmunoprecipitation with the indicated affinity-purified antibodies. Total (20%), nonbound (20%), and bound fractions (100%) were analyzed by western blotting with the indicated antibodies. PI, preimmune serum. (B) As in (A) with the difference that antibodies to Dbi1 were used for coimmunoprecipitation. (C) Wild type and mitochondria isolated from *tim23G112L* mutant cells were treated and analyzed as in (A) with the difference that only total (20%), and bound (100%) fractions were analyzed. (D) Wild type and mitochondria isolated from Tim23-depleted cells were treated and analyzed as in (C). (E) As in (D) with the difference that total (T, 20%) and nonbound (20%) fractions were analyzed. (F) 10, 20, and 50 μg of isolated mitochondria (protein amount) were dissolved in Laemmli buffer and analyzed by western blotting. Source data are available online for this figure.

the other hand, Tim17 has a disulfide bond in the IMS close to the IM that is not present in Tim23. Inspection of the Dbi1 sequence revealed a conserved CxxC motif on the IMS side of the protein, preceding TM1. This motif reminded us of the redox active CPC motif of Mia40 and we thus decided to investigate the redox state of the two cysteines in Dbi1. To this end, we incubated isolated mitochondria with alkylating reagent methyl-polyethylene glycol-maleimide (mmPEG$_{24}$) which reacts with reduced, but not oxidized, cysteine residues leading to a characteristic ca. 2 kDa shift per free thiol that can be detected by SDS-PAGE (Ramesh et al, 2016). mmPEG$_{24}$ was omitted from one sample which subsequently served as a nonmodified control and one sample was pretreated with a strong reducing reagent TCEP at 96 °C prior to incubation with mmPEG$_{24}$, to reduce all potentially present disulfide bonds, so that all cysteine residues can be modified. The characteristic shift in Dbi1 was only detectable when the sample was reduced prior to labeling (Fig. 3B), demonstrating that the two cysteine residues in Dbi1 form an intramolecular disulfide bond under steady state. Thus, Dbi1 could in principle serve as an oxidoreductase. We then analyzed the oxidation state of Tim17 in WT and in mitochondria lacking Dbi1. In WT mitochondria, we observed that two cysteine residues are available for labeling, whereas the other two are only accessible upon prior reduction of the protein (Fig. 3C). This finding is consistent with the previous reports that C10, N-terminal to TM1, and C77, C-terminal to TM2, of Tim17 form the disulfide bond, whereas C118 and C120, present in TM4, are in the reduced state (Ramesh et al, 2016; Wrobel et al, 2016). In mitochondria lacking Dbi1 about half of Tim17 was

present in the fully reduced state (Fig. 3C). In contrast, the oxidation states of Mia40 and Erv1, essential components of the mitochondrial disulfide relay system, and of Tim13, one of its substrates, were unaffected by the absence of Dbi1. The fraction of reduced Tim17 was even larger when its oxidation state was analyzed directly in cells lacking Dbi1 (Fig. EV2A), suggesting that some of Tim17 may get re-oxidized during isolation of mitochondria. These results show that Dbi1 influences the disulfide bond formation in Tim17 in a manner independent of the disulfide relay system. To analyze the role of the CxxC motif in Dbi1 in the oxidation of Tim17, we mutated the two cysteine residues, individually and together, to serine residues. Mutation of C27 of Dbi1 was sufficient to produce the same effect on the oxidation state of Tim17 as the deletion of the whole protein (Fig. 3D). Again, oxidation states of Mia40, Erv1 and Tim13 were not affected in any of the Dbi1 mutants (Fig. EV2B). Furthermore, mutations of the two cysteine residues which form the disulfide bond in Tim17 did not affect the oxidation state of Dbi1 (Fig. EV2C), demonstrating that Tim17 is a substrate of Dbi1 and not vice versa. These cysteine residues in Tim17 were, however, essential for interaction of Tim17 with Dbi1 as Tim17 did not associate with Dbi1 in *tim17C10S/C77S* mutant (Fig. 3E). Furthermore, in cells lacking Dbi1 and in *tim17C10S/C77S* mutant, Tim17 was degraded faster compared to WT (Fig. 3F), showing that the disulfide bond stabilizes the protein and prevents its degradation.

The results presented so far suggest that Dbi1 is an oxidoreductase that introduces the disulfide bond into Tim17 and chaperones its assembly with Tim23. To support the notion that

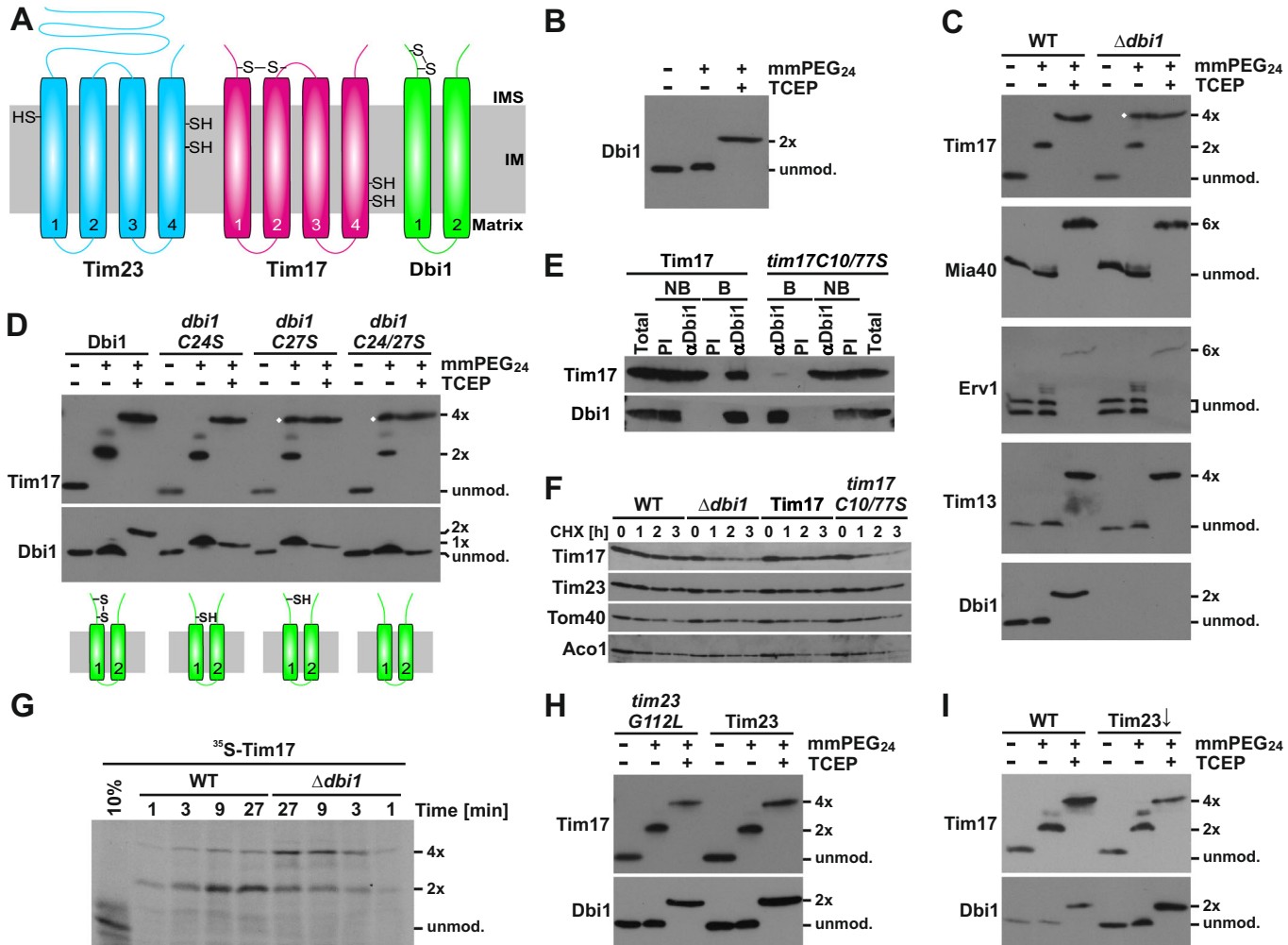

**Figure 3. Dbi1 is involved in the formation of Tim17 disulfide bond.**

(A) Schematic representations of Tim23, Tim17, and Dbi1 and their cysteine residues. (B) Isolated wild type mitochondria were solubilized in SDS-containing buffer and incubated with the free thiol reactive reagent methyl-polyethylene glycol-maleimide (mmPEG$_{24}$). One sample was fully reduced with TCEP at 96 °C prior to incubation with mmPEG$_{24}$. Samples were analyzed by western blotting. (C) Wild type (WT) and mitochondria lacking Dbi1 (Δdbi1) were treated and analyzed as in (B). (D) Mitochondria containing indicated Dbi1 variants were treated and analyzed as in (B). (E) Isolated mitochondria were solubilized with digitonin and subjected to coimmunoprecipitation using affinity-purified antibodies to Dbi1. Antibodies from the preimmune serum (PI) served as a negative control. Total (20%), nonbound (NB, 20%) and bound (B, 100%) fractions were analyzed by western blotting. (F) Cycloheximide (CHX) was added to the cultures growing in YPD medium at 37 °C. Samples were taken at the indicated time points, whole cell extracts were prepared and analyzed by western blotting. (G) $^{35}$S-labeled Tim17 was imported into isolated wild type and mitochondria lacking Dbi1 (Δdbi1). At the indicated time points, samples were removed, treated with proteinase K to remove all nonimported material and subsequently labeled with mmPEG$_{24}$. Samples were analyzed by SDS-PAGE and autoradiography. (H) Thiol labeling of mitochondria isolated from wild type and tim23G112L mutant cells was done as in (B). (I) Wild type and mitochondria isolated from Tim23-depleted cells were subjected to thiol labeling as in (B). White diamonds highlight the fully reduced Tim17 in mutant mitochondria. Source data are available online for this figure.

Dbi1 acts early during the biogenesis of Tim17, we imported $^{35}$S-labeled Tim17 into WT and mitochondria lacking Dbi1. Whereas the disulfide bond in Tim17 was promptly formed in WT, in mitochondria lacking Dbi1 about half of newly imported Tim17 remained in the fully reduced state (Fig. 3G), confirming the role of Dbi1 early in the biogenesis of Tim17. Since we observed an increased interaction between Dbi1 and Tim17 in mitochondria with destabilized and/or absent Tim17–Tim23 core complex, we wondered whether Dbi1 introduces the disulfide bond into Tim17 upon its binding to Tim23 or before. To this end, we analyzed the oxidation state of Tim17 in tim23G112L mutant mitochondria and in mitochondria lacking Tim23. In both cases, the disulfide bond in

Tim17 was fully formed (Fig. 3H,I), suggesting that the disulfide bond formation normally precedes assembly of Tim17 into the TIM23 complex. This finding also shows that Dbi1 remains bound to Tim17 even after the oxidation step took place, further supporting the notion that Dbi1 has a pronounced chaperone function as well.

## Tim22 is also a substrate of Dbi1

Tim22 is the central component of the carrier translocase and, like Tim17, contains the disulfide bond which locks its TMs 1 and 2 in a stable conformation (Fig. 4A). Considering structural similarities

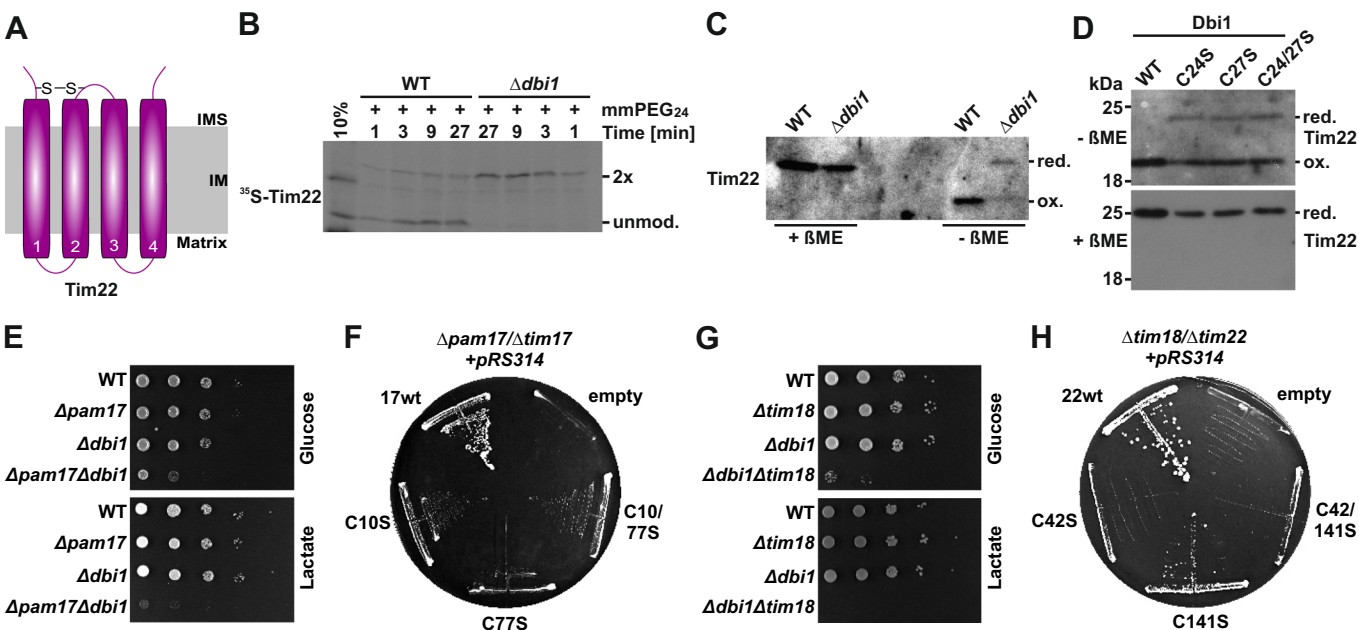

**Figure 4. Tim22 is a further substrate of Dbi1.**

(A) Cartoon representing Tim22 and its cysteine residues. (B) $^{35}$S-labeled Tim22 was imported into isolated wild type and mitochondria lacking Dbi1. At the indicated time points, samples were removed, treated with proteinase K to remove all nonimported material and then labeled with mmPEG$_{24}$. Samples were analyzed by SDS-PAGE followed by autoradiography. (C) Wild type and mitochondria lacking Dbi1 were dissolved in reducing (+ßME) or nonreducing (-ßME) Laemmli buffer and analyzed by western blotting. (D) Mitochondria containing indicated variants of Dbi1 were treated and analyzed as in (C). (E) Serial dilutions of indicated cells were grown on glucose and lactate medium at 30 °C. (F) Tim17 shuffling strain, in the background of *PAM17* deletion, was transformed with plasmids expressing indicated variants of Tim17 and transformants were plated on medium containing 5'-fluoroorotic acid (5'FOA). Plasmid expressing the wild type version of Tim17 served as a positive and an empty plasmid as a negative control. (G) Serial dilutions of indicated cells were grown on glucose and lactate medium at 30 °C. (H) Tim22-shuffling strain, in the background of *TIM18* deletion, was transformed with plasmids expressing indicated variants of Tim22 and cells were plated on medium containing 5'FOA. Plasmid expressing the wild type version of Tim22 served as a positive and an empty plasmid as a negative control. Source data are available online for this figure.

between the two proteins, we wondered whether Tim22 could also be a substrate of Dbi1. When we imported $^{35}$S-labeled Tim22 into WT and mitochondria lacking Dbi1, we observed that the oxidation of newly imported Tim22 was essentially absent in mitochondria lacking Dbi1 (Fig. 4B). Similarly, endogenous Tim22 was present in a reduced state both in mitochondria lacking Dbi1 (Fig. 4C) and in the ones containing mutations in its CxxC motif (Fig. 4D). In contrast, absence of the disulfide bond in Tim22 had no influence on the oxidation state of Dbi1, showing that Tim22, like Tim17, is a substrate of Dbi1 and not vice versa (Fig. EV3A,B). Importantly, the absence of the disulfide bond in Tim22 also had no influence on the oxidation state of Tim17 (Fig. EV3B), excluding potential secondary effects of Tim22 on Tim17.

The disulfide bonds in either Tim17 or Tim22 are not essential for yeast cell viability, however, they influence the stabilities of the respective proteins and affect protein translocation along the respective import pathways, especially under stress conditions (Okamoto et al, 2014; Ramesh et al, 2016). We thus reasoned that Dbi1 should influence both major mitochondrial protein transloca-tion pathways. To this end, we analyzed the growth of cells lacking Dbi1, alone or in combination with one nonessential subunit of either TIM23 or TIM22 complexes. Whereas deletion of either *PAM17* or *DBI1* alone had no effect on the growth of yeast cells on either fermentable (glucose) or nonfermentable (lactate) medium,

containing medium and even worse on lactate-containing medium (Fig. 4E). If the strong growth defect of Δ*pam17*/Δ*dbi1* cells was due to the absence of the disulfide bond in Tim17, then deletion of *PAM17* should show the same effect in combination with Tim17 cysteine mutants. This was indeed the case (Fig. 4F). For the TIM22 complex, we chose Tim18 as a nonessential subunit. Also, in this case, simultaneous deletion of *DBI1* and *TIM18* dramatically impaired the growth of yeast cells. The double mutant grew very poorly on fermentable medium and even worse on nonfermentable medium (Fig. 4G). Similarly, deletion of *TIM18* combined with the cysteine mutants of Tim22 resulted in a very strong impairment of the growth of yeast cells (Fig. 4H). Taken together, Dbi1 influences translocation along both the presequence and carrier pathways through disulfide bond formation and thus stabilization of the central components of the two translocases.

## Substrates of Dbi1 are not limited to translocases components

Data presented so far suggest that Dbi1 is involved in the formation of disulfide bonds that connect and thus stabilize the conformation of two transmembrane helices in its substrates. We wondered whether we could identify additional proteins with such a motif. Based on the recent reports that implicated Dbi1 in processes related to mtDNA and there encoded proteins

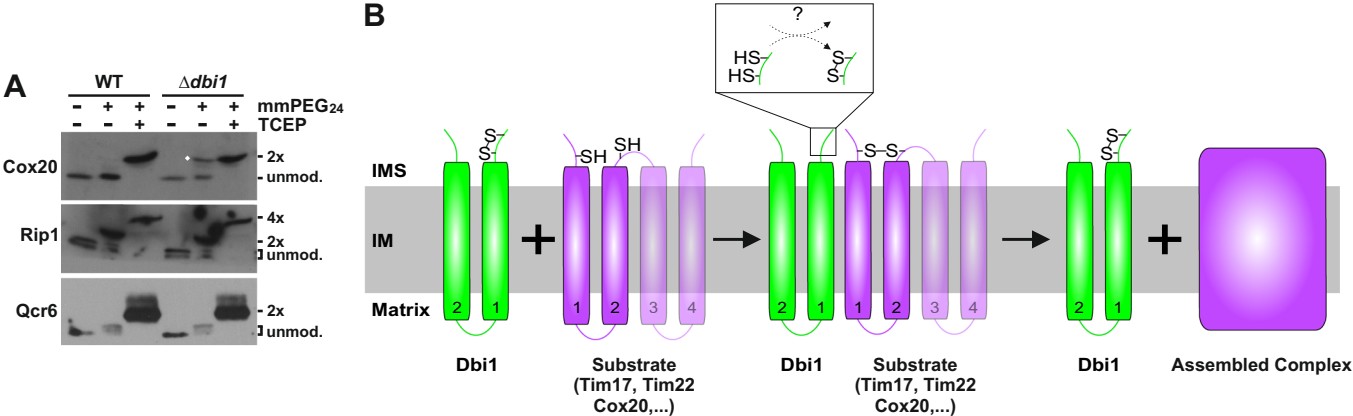

**Figure 5. Role of Dbi1 in oxidation and assembly of inner membrane proteins.**

(A) Isolated mitochondria were solubilized in SDS-containing buffer and incubated with mmPEG$_{24}$, where indicated. One sample was fully reduced with TCEP at 96 °C prior to labeling. Samples were analyzed by SDS-PAGE followed by western blotting using indicated antibodies. White diamond highlights the fully reduced Cox20 in mitochondria lacking Dbi1. (B) Model of Dbi1 function. See text for details. Source data are available online for this figure.

(Dickinson et al, 2022; Merz and Westermann, 2009; Soares et al, 2024), we looked for a potential candidate. Cox20 drew our attention not only because it is a chaperone involved in biogenesis of Cox2 (Hell et al, 2000) but also because it contains two transmembrane domains flanked by two conserved cysteine residues (Riemer et al, 2009; Szklarczyk et al, 2013), fitting the putative motif of Dbi1 substrates. Cox20 was indeed present in the oxidized state in WT mitochondria whereas approximately half of it was in the reduced state in mitochondria lacking Dbi1 (Fig. 5A). In contrast, oxidation states of Rip1 and Qcr6, that both contain disulfide bonds in the soluble domains in IMS, were not affected in mitochondria lacking Dbi1. Thus, Cox20 is another substrate of Dbi1. Structural predictions of Dbi1 in complex with Tim17, Tim22, and Cox20, made by AlphaFold3 (Abramson et al, 2024), suggest that its CxxC motif is positioned at the IMS-IM border, atop the cysteine residues of its substrates (Fig. EV4A–C).

## Concluding remarks

With the identification of Dbi1 here, an additional oxidoreductase was discovered, indicating that the formation of disulfide bonds in the IMS can take multiple routes. In this way, the situation in IMS starts to resemble the endoplasmic reticulum where various protein disulfide isomerases are known and are responsible for oxidation of different substrates (Herrmann and Riemer, 2014). The data presented here suggest that Dbi1 operates independently of the disulfide relay system. This is further supported by the fact that all currently known substrates of the disulfide relay system are relatively small, soluble IMS proteins. In contrast, Dbi1 appears to specialize in oxidation of inner membrane proteins. It will be interesting to identify the whole spectrum of Dbi1 substrates using the knowledge of the putative motif consisting of two transmembrane segments flanked by cysteine residues. Association of DMAC1 with assembly intermediates of complex I may suggest the presence of a substrate there, though the oxidoreductase function of DMAC1 was not analyzed so far. NDUFA11, which belongs to the same protein family as Tim17, Tim22 and Tim23

(Zarsky and Dolezal, 2016), appears to be an obvious candidate. It also remains to be determined in the future how Dbi1 gets its disulfide bond. Our data suggest that Mia40, Erv1 and Dbi1 itself are not involved in the process (Fig. EV4B–D), hinting at the presence of another sulfhydryl oxidase in the IMS. Since both Tim17 and Dbi1 are fully oxidized in their complex, the putative sulfhydryl oxidase appears to be able to reoxidize Dbi1 while it is still bound to its substrate (Fig. 5B). It is thus tempting to speculate that this putative enzyme may also be embedded in the IM.

In the last ca. 15 years it became clear that the presequence pathway is the main sensor of the mitochondrial well-being and that many stress-induced rescue/degradation pathways, such as mitophagy, mtUPR, and ISR, are all initiated by the impaired import of specific substrates of the presequence pathway (Song et al, 2021). It is tempting to speculate that the redox state of mitochondria may also be sensed by the presequence pathway. In this respect, it is interesting to note that two isoforms of Tim17, Tim17A, and Tim17B, are present in humans (Bauer et al, 1999). The presence of the disulfide bond was previously reported in Tim17B (Wrobel et al, 2016). The respective cysteine residues are also conserved in Tim17A, however, their oxidation state has not been analyzed so far. What was however previously shown is that Tim17A is degraded under stress conditions (Rainbolt et al, 2013). If the disulfide bond is indeed present in Tim17A, it is possible that this disulfide bond is redox-regulated, allowing for rapid turnover of the protein and thus adaptation of the import capacity of mitochondria to the stress conditions (Mokranjac, 2016).

Based on the available data, we propose the following model of the function of Dbi1 (Fig. 5B). Substrates of Dbi1 are inner membrane proteins with at least two transmembrane segments flanked by cysteine residues. Dbi1 recognizes its substrates through their specific cysteine residues, introduces the disulfide bond, and then chaperones their further assembly into the respective complexes. If incorporation of newly imported and oxidized substrates into their respective complexes is not possible, they remain bound to Dbi1 in a substrate-chaperone complex.

# Methods

### Reagents and tools table

| Reagent/resource | Reference or source | Identifier or catalog number |
|---|---|---|
| *Yeast strains* | | |
| YPH499 | (Sikorski & Hieter, 1989) | |
| YPH499 *Δtim23::KAN* + pRS314-prom-Tim23-flank | (Gunsel et al, 2020) | |
| YPH499 *Δtim23::KAN* + pRS314-prom-Tim23_87A5-flank | (Gunsel et al, 2020) | |
| YPH499 *Δtim23::KAN* + pRS314-prom-Tim23-flank *Dbi1-His₆::HIS3* | This study | |
| YPH499 *Δdbi1::HIS3* | This study | |
| YPH499 *GAL10*-TIM23 | (Popov-Celeketic et al, 2008) | |
| YPH499 *Δtim23::KAN* + pRS315-prom-Tim23-flank | (Demishtein-Zohary et al, 2015) | |
| YPH499 *Δtim23::KAN* + pRS315-prom-Tim23_G112L-flank | (Demishtein-Zohary et al, 2015) | |
| YPH499 *Δdbi1::HIS3* + pRS314_prom-Dbi1-flank | This study | |
| YPH499 *Δdbi1::HIS3* + pRS314_prom-Dbi1_C24S-flank | This study | |
| YPH499 *Δdbi1::HIS3* + pRS314_prom-Dbi1_C27S-flank | This study | |
| YPH499 *Δdbi1::HIS3* + pRS314_prom-Dbi1_C24/27S-flank | This study | |
| YPH499 *Δtim17::HIS3* + pRS314_prom-Tim17-flank | This study | |
| YPH499 *Δtim17::HIS3* + pRS314_prom-Tim17_C10S-flank | This study | |
| YPH499 *Δtim17::HIS3* + pRS314_prom-Tim17_C77S-flank | This study | |
| YPH499 *Δtim17::HIS3* + pRS314_prom-Tim17_C10/77S-flank | This study | |
| YPH499 *Δtim22::HIS3* + pRS314_prom-Tim22-flank | This study | |
| YPH499 *Δtim22::HIS3* + pRS314_prom-Tim22_C42S-flank | This study | |
| YPH499 *Δtim22::HIS3* + pRS314_prom-Tim22_C141S-flank | This study | |
| YPH499 *Δtim22::HIS3* + pRS314_prom-Tim22_C42/141S-flank | This study | |
| YPH499 *Δpam17::HIS3* | (Popov-Celeketic et al, 2008) | |
| YPH499 *Δpam17::HIS3 Δdbi1::KAN* | This study | |
| YPH499 *Δpam17::HIS3 Δtim17::KAN* + pVT-102U-Tim17 + pRS314 | This study | |
| YPH499 *Δpam17::KAN Δtim17::HIS3*+ pVT-102U-Tim17 + pRS314_prom-Tim17-flank | This study | |
| YPH499 *Δpam17::KAN, Δtim17::HIS3* + pVT-102U-Tim17 + pRS314_prom-Tim17_C10S-flank | This study | |
| YPH499 *Δpam17::KAN, Δtim17::HIS3* + pVT-102U-Tim17 + pRS314_prom-Tim17_C77S-flank | This study | |
| YPH499 *Δpam17::KAN Δtim17::HIS3* + pVT-102U-Tim17 + pRS314_prom-Tim17_C10/77S-flank | This study | |
| YPH499 *Δtim18::HYG Δtim22::HIS3* + pVT-102U-Tim22 + pRS314 | This study | |
| YPH499 *Δtim18::HYG Δtim22::HIS3* + pVT-102U-Tim22 + pRS314_prom-Tim22-flank | This study | |
| YPH499 *Δtim18::HYG Δtim22::HIS3* + pVT-102U-Tim22 + pRS314_prom-Tim22_C42S-flank | This study | |
| YPH499 *Δtim18::HYG Δtim22::HIS3* + pVT-102U-Tim22 + pRS314_prom-Tim22_C141S-flank | This study | |
| YPH499 *Δtim18::HYG Δtim22::HIS3* + pVT-102U-Tim22 + pRS314_prom-Tim22_C42/141S-flank | This study | |
| YPH499 *Δtim18::HYG* | This study | |
| YPH499 *Δtim18::HYG Δdbi1::HIS3* | This study | |
| YPH499 *GAL10-MIA40* | (Terziyska et al, 2005) | |
| YPH499 *GAL10-ERV1* | (Mesecke et al, 2005) | |
| *Recombinant DNA (plasmids)* | | |
| pRS314 | (Sikorski & Hieter, 1989) | |
| pRS314-prom-Tim23-flank | (Gunsel et al, 2020) | |
| pRS314-prom-Tim23_87A5-flank | (Gunsel et al, 2020) | |
| pRS315-prom-Tim23-flank | (Demishtein-Zohary et al, 2015) | |
| pRS315-prom-Tim23_G112L-flank | (Demishtein-Zohary et al, 2015) | |
| pRS314_prom-Dbi1-flank | This study | |
| pRS314_prom-Dbi1_C24S-flank | This study | |
| pRS314_prom-Dbi1_C27S-flank | This study | |

| Reagent/resource | Reference or source | Identifier or catalog number |
|---|---|---|
| pRS314_prom-Dbi1_C24/27S-flank | This study | |
| pRS314_prom-Tim17-flank | (Demishtein-Zohary et al, 2017) | |
| pRS314_prom-Tim17_C10S-flank | This study | |
| pRS314_prom-Tim17_C77S-flank | This study | |
| pRS314_prom-Tim17_C10/77S-flank | This study | |
| pRS314_prom-Tim22-flank | This study | |
| pRS314_prom-Tim22_C42S-flank | This study | |
| pRS314_prom-Tim22_C141S-flank | This study | |
| pRS314_prom-Tim22_C42/141S-flank | This study | |
| pVT-102U-Tim17 | (Demishtein-Zohary et al, 2017) | |
| pVT-102U-Tim22 | This study | |
| pVT-102U-Dbi1 | This study | |
| pGEM4-Dbi1 | This study | |
| pGEM4-Tim17 | This study | |
| pGEM4-Tim22 | This study | |
| *Antibodies* | | |
| Cox2 | (Herrmann et al, 1995) | 20784 |
| Cox20 | (Hell et al, 2000) Kind gift from A. Tzagoloff | N/A |
| Dbi1 | This study | YDL Cpep affi |
| Erv1 | (Mesecke et al, 2005) | 189 |
| Mdj1 | Neupert Lab antibodies | 548 |
| Mia40 | (Terziyska et al, 2005) | Mia40 Cpep |
| Pam17 | (Popov-Celeketic et al, 2008) | 378 affi |
| Phb2 | (Steglich et al, 1999) | 118 |
| Porin | Neupert Lab antibodies | 87118 |
| Qcr6 | (Cruciat et al, 1999) | 75 |
| Rip1 | (Wagener et al, 2011) | Rip1 Cpep |
| Ssc1 | (Sichting et al, 2005) | 347 affi |
| Tim13 | (Lutz et al, 2003) | 369 |
| Tim14 | This study | 445 |
| Tim17 | Neupert Lab antibodies | Tim17Cpep affi |
| Tim22 | Kind gift from N. Pfanner | N/A |
| Tim23N | Neupert Lab antibodies | Tim23Npep affi |
| Tim23C | Neupert Lab antibodies | Tim23Cpep affi |
| Tim44 | (Banerjee et al, 2015) | 388 affi |
| Tim50 | (Mokranjac et al, 2003a) | 293 affi |

| Reagent/resource | Reference or source | Identifier or catalog number |
|---|---|---|
| Tom40 | (Genge et al, 2023) | 547 affi |
| Tom70 | (Stan et al, 2000) | 312 |
| Yme1 | (Schreiner et al, 2012) | Yme1Cpep |
| *Oligonucleotides and other sequence-based reagents* | | |
| Primers | 5′-3′ | |
| Dbi1Δfw | GAA ACA AAA CAC CGC TTA GAG AGA AAC AGC AAG TGG TGA AAA ACA CGT ACG CTG CAG GTC GAC | |
| Dbi1Δrv | TTA TGT ACA GAA AAT TTC ACG ATT GAG AGA AGCTTG CGA ATA ATA ATC GAT GAA TTC GAG CTC G | |
| SacpDbi1 | CCC GAG CTC TAG ATT TTT CCA ACA C | |
| Dbi1fXho | GGG CTC GAG TTA GTT AGG AAC ATA TTG CAC | |
| BamDbi1 | CCC GGA TCC ATG AGT AAT ATT TTG GCA GTG | |
| Dbi1Hind | GGG AAG CTT CTA GTT AGC CTG TGT TTC ACC | |
| Dbi1Hisfw | AAG AAC TAT AAG AAA TTA AGT AAC GAT GGT GAA ACA CAG GCT AAC CAT CAC CAT CAC CAT CAC | |
| Dbi1_C24S_fw | GAT TCC GTT CCT TGT CAA GTC ATG AGC ACT ATG | |
| Dbi1_C27S_ fw | GAT TGC GTT CCT TCT CAA GTC ATG AGC ACT ATG | |
| Dbi1_C24/27S_fw | GAT TCC GTT CCT TCT CAA GTC ATG AGC ACT ATG | |
| Dbi1_Cysmut_rv | CAT AGT CTC TTC TTT TTC GAG CTC CCT TTG TGG | |
| EcoTim17 | CCC GAA TTC ATG TCA GCC GAT CAT TCG AG | |
| Tim17Pst | GGG CTG CAG CTA AGC TTG CAG AGG TTG AG | |
| Tim17_C10S_fw | CCA TCT CCT ATA GTC ATA CTA AAT GAT TTC | |
| Tim17_C10S_rv | ATC CCT CGA ATG ATC GGC TGA CAT AAC GC | |
| Tim17_C77S _fw | GAT TCC GCT GTG AAG GCC GTT AGA AAA G | |
| Tim17_C77S_rv | AAA CGT TGA AAA TAA ACC ACC CCA CAC | |
| Pam17Δfw | AAG AAG TGT TAA AAA CAT TCA GAA AAC ATT GTC CGC CTC TTC AAA CGT ACG CTG | |
| Pam17Δrv | GTA TAT ATA CAG AGT CTG AGA AGA AGG AAA AGA TCA CAC GTT CAA ATC GAT GAA | |
| Tim22Δfw | GCC AAA TAA GAT CAA GAA ATT GTG ATT TTA AAT ACT TTA TAC GAA GAT GCG TAC GCT GCA GGT CGA C | |

| Reagent/resource | Reference or source | Identifier or catalog number |
|---|---|---|
| Tim22Δrv | GGT GAT ATT TAC AAA TAT AAA ACA TTC ATC GTT CGT CGA AAT TGG CTA TTC AAT CGA TGA ATT CGA GCTC G | |
| SacpTim22 | CCC GAG CTC GAT GAG GGC AAT GAT ATG TTA | |
| Tim22fXho | GGG CTC GAG TGA TCG GCC CAG AGT CTA G | |
| BamTim22 | CCC GGA TCC ATG GTG TAC ACA GGA TTT GG | |
| Tim22Pst | GGG CTG CAG TCA TTC TTT AAA ATC GTT TTG AGG | |
| Tim22_C42S_fw | GGG AAA TCA GTT GTG AGT GGT GTA ACA G | |
| Tim22_C42S_rv | GGG TGA GGA AGT CAT GAA GTT CAT GAT C | |
| Tim22_C141S_fw | AGT CAG TCA TAG ATA TCATAC CAA TG | |
| Tim22_C141S_rv | CCA CGC CGG CAT ATA TCA TAC CAA TG | |
| Tim18Δfw | CGG TGA TGC GAG GTG CAA CAA CTG AGT AAT TTA ATA CCT TTG GTA CGT ACG CTG CAG GTC GAC | |
| Tim18Δrv | AAT GAA ATC TTA GAA ATG CAA AAA AAA AGA AAA AGT ATG GGT GAG ATC GAT GAA TTC GAGCTC | |
| *Chemicals, enzymes and other reagents* | | |
| [35]S-Methionine | Revvity Health Sciences | 06174/1 |
| 5-Fluoroorotic acid (5-FOA) | US Biological Life Sciences | F5050 |
| Adenosine-5′-triphosphate (ATP) | Roche Diagnostics | 10519979001 |
| Bacto Peptone Gibco | Otto Nordwald | 211830 |
| Bacto Yeast extract Gibco | Otto Nordwald | 212730 |
| Creatine kinase (CK) | Roche Diagnostics | 05168546188 |
| Creatine phosphate (CP) | Sigma-Aldrich | 237911 |
| Cycloheximide (CHX) | Sigma-Aldrich | 01810 |
| Digitonin | Calbiochem | 300410 |
| Dimethylsulfoxide (DMSO) | Sigma-Aldrich | 20139 |
| Disuccinimidyl glutarate (DSG) | Thermo Scientific | PI20593 |
| Methyl-polyethylene glycol$_{24}$-maleimide (mmPEG$_{24}$) | Thermo Scientific | 22713 |
| Sodium carbonate (Na$_2$CO$_3$) | Carl-Roth | 8653.1 |
| Nicotinamide adenine dinucleotide (NADH) | GERBU | 1051 |
| Ni-NTA-Agarose | Qiagen | 30210 |
| Protein A Sepharose CL-4B | Cytiva | 17096303 |
| Proteinase K (PK) | Roche Diagnostics | 3115836001 |
| TNT® Coupled Reticulocyte Lysate | Promega | L4600 |

| Reagent/resource | Reference or source | Identifier or catalog number |
|---|---|---|
| Trichloroacetic acid (TCA) | Sigma-Aldrich | T6399 |
| Tris(2-carboxyethyl) phosphine (TCEP) | Sigma-Aldrich | C4706 |
| Triton-X100 | Sigma-Aldrich | X100 |
| Yeast Nitrogen Base | Otto Nordwald | 291920 |
| *Software* | | |
| Fiji 2.0.0 | | |
| PyMOL 2.5.2 | | |
| HMMER | https://www.ebi.ac.uk/Tools/hmmer/search/jackhmmer (Potter et al, 2018) | |
| AlphaFold3 | https://golgi.sandbox.google.com/ (Abramson et al, 2024) | |
| TheCellMap | https://thecellmap.org/ (Usaj et al, 2017) | |
| *Other* | | |
| NuPAGE 4-12% Bis-Tris Gel | Thermo Fisher Scientific | NP0322BOX |

## Yeast strains and plasmids

All yeast strains used in this study are based on *Saccharomyces cerevisiae* YPH499 wild type strain (Mat a, *ade2-101, his3-Δ200, leu2-Δ1, ura3-52, trp1-Δ63, lys2-801*) (Sikorski and Hieter, 1989). *PAM17* deletion strain (Popov-Celeketic et al, 2008), Tim17 shuffling strain (Banerjee et al, 2015), *GAL-MIA40* (Terziyska et al, 2005), *GAL-ERV1* (Mesecke et al, 2005) and Tim23 strains *tim23-87A5* (Gunsel et al, 2020), *tim23G112L* (Demishtein-Zohary et al, 2015) and *GAL-TIM23* (Popov-Celeketic et al, 2008) were previously described. Chromosomal tagging and deletions were done by homologous recombination of PCR products, as described previously (Knop et al, 1999). Dbi1- and Tim22-shuffling strains were made by transforming wild type versions of Dbi1 and Tim22, respectively, expressed from pVT-102U plasmids (Vernet et al, 1987), into YPH499 strain and subsequently deleting chromosomal copies of the respective genes. All strains were confirmed by PCRs both for the correct insertion of the deletion cassette and for the absence of the WT allele. For complementation analyses, ORFs, including their endogenous promoters and 3′UTRs, were cloned into pRS314 or pRS315 centromeric yeast plasmids (Sikorski and Hieter, 1989). Point mutations were introduced by side-directed mutagenesis PCRs. All constructs were confirmed by DNA sequencing. Plasmids were transformed into respective shuffling strains and selected on selective glucose media lacking appropriate markers. An empty plasmid and a plasmid expressing WT version of the protein were used as negative and positive controls, respectively. Cells that lost WT copies of the genes on *URA3* plasmids were selected on medium containing 5′-fluoroorotic acid at 30 °C.

Yeast cells were typically grown in full or selective lactate medium containing 0.1% glucose at 30 °C except for the *tim23G112L* mutant which was grown in YPD medium at 24 °C. Mitochondria were isolated by differential centrifugation (Genge

et al, 2023) from cells in logarithmic growth phase. Control strains were always grown in parallel under identical conditions.

## Crosslinking

Isolated yeast mitochondria were resuspended in SI buffer (50 mM HEPES-KOH, 0.6 M Sorbitol, 80 mM KCl, 100 mM Mg(Ac)$_2$, 2 mM EDTA, 2.5 mM MnCl$_2$, pH 7.2) and energized with 4 mM ATP, 5 mM NADH, 10 mM creatine phosphate (CP) and 0.1 mg/mL creatine kinase (CK) for 3 min at 25 °C. Disuccinimidyl glutarate (DSG) was added to samples for crosslinking from a freshly prepared 100-fold stock in DMSO, while DMSO was added to the negative controls. The samples were incubated for 30 min on ice. Excess crosslinker was quenched by addition of 100 mM glycine, pH 8.8. After a 10 min incubation on ice, samples were diluted with SH-buffer (0.6 M sorbitol, 20 mM HEPES-KOH, pH 7.4) and mitochondria were reisolated by centrifugation (10 min, 18,000 × g, 4 °C). The pellets were resuspended in 2× Laemmli buffer containing 3% β-mercaptoethanol, boiled for 5 min at 95 °C, and analyzed via SDS-PAGE and immunodecoration with Tim23N antibodies. For analyzing crosslinking adducts via binding to NiNTA-agarose beads, after crosslinking, mitochondria were resuspended in solubilization buffer (50 mM NaH$_2$PO$_4$, 100 mM NaCl, 10 mM imidazole, pH 8.0, 1% (w/v) SDS, 2 mM PMSF, added just prior to use), incubated for 5 min at RT and subsequently diluted with cold washing buffer (50 mM NaH$_2$PO$_4$, 100 mM NaCl, 10 mM imidazole, pH 8.0, 0.2% (v/v) Triton X-100, 2 mM PMSF, added just prior to use). The supernatant after centrifugation (20 min, 124,500 × g, 2 °C) was incubated with NiNTA beads for 30 min on a rotating platform at 4 °C. The supernatant was discarded and the beads were washed three times for 5 min with 200 µL washing buffer. Specifically bound proteins were eluted with 2× Laemmli buffer containing 3% β-mercaptoethanol and 300 mM imidazole by incubation for 5 min at 95 °C.

## Submitochondrial localization

Isolated yeast mitochondria were incubated for 20 min on ice with 3 mM ATP in either SH-buffer (0.6 M sorbitol, 20 mM HEPES-KOH pH 7.4) (Mitochondria), 20 mM HEPES-KOH (Mitoplasts) or 20 mM HEPES-KOH with 0.25% Triton X-100 (Triton X-100). Proteinase K (PK) was added where indicated. Protease digestion was stopped by the addition of 1 mM PMSF and further incubation for 5 min on ice. Mitochondria and mitoplasts were reisolated (10 min, 18,000 × g, 4 °C), resuspended in 2× Laemmli buffer with 3% β-mercaptoethanol, incubated for 5 min at 95 °C and stored at −20 °C until loaded on the gel. The Triton-solubilized sample was TCA-precipitated before resuspension in 2× Laemmli buffer with 3% β-mercaptoethanol.

For the carbonate extraction, isolated mitochondria were mixed with 20 mM HEPES-KOH, pH 7.4 and 200 mM freshly prepared Na$_2$CO$_3$ and incubated on a rotating platform at 4 °C for 30 min. One aliquot (total, T) was directly TCA-precipitated and the other was centrifuged (30 min, 186,000 × g, 2 °C). The supernatant of this centrifugation (S), which contains soluble and peripheral membrane proteins, was first TCA-precipitated while the pellet (P), which contains integral membrane proteins, was directly resuspended in 2× Laemmli buffer with 3% β-mercaptoethanol and incubated for 5 min at 95 °C.

## Coimmunoprecipitation

Protein A-Sepharose CL-4B beads were incubated with affinity-purified antibodies for 1 h on a rotating platform at 4 °C, washed and equilibrated with the solubilization buffer (20 mM Tris-HCl, 80 mM KCl, 10% glycerol, 5 mM EDTA, pH 8.0) containing 0.05% (w/v) digitonin. Isolated mitochondria were solubilized with solubilization buffer containing 1% digitonin and 2 mM PMSF for 15 min at 4 °C. After a clarifying spin, the supernatant was added to the beads and rotated for 45 min at 4 °C. Nonbound material was collected. Beads were washed three times with solubilization buffer containing 0.05% (w/v) digitonin. Bound fractions were eluted with 2× Laemmli buffer with 3% β-mercaptoethanol, incubated for 3 min at 95 °C, and analyzed via SDS-PAGE and immunodecoration. When Tim50 was analyzed β-mercaptoethanol was omitted from the Laemmli buffer.

## Thiol modification assay

The oxidation state of proteins in isolated mitochondria was determined by their accessibility to the alkylating reagent methyl-polyethylene glycol-maleimide (mmPEG$_{24}$), essentially as described by Ramesh et al (Ramesh et al, 2016). Briefly, isolated mitochondria were resuspended in AMS buffer (80 mM Tris, pH 7, 10% glycerol, 2% SDS, 0.05% Bromocresol green) and either incubated with DMSO or 15 mM mmPEG$_{24}$ for 1 h in the dark at RT. When indicated, the samples were pretreated with 10 mM TCEP and incubated for 20 min at 96 °C prior to the addition of mmPEG$_{24}$. Samples were incubated for 1 min at 95 °C and analyzed via SDS-PAGE and immunodecoration.

To assess the oxidation state of proteins in cells, the protocol described by Leichert and Jakob (Leichert and Jakob, 2004) was modified. Briefly, yeast cells were grown to log phase, harvested by centrifugation, washed with water and the resulting cell pellets resuspended in 12% TCA by vortexing. After ON incubation at −80 °C, the cells were reisolated by centrifugation (18,000 × g for 20 min at 4 °C), washed with 5% TCA and finally resuspended in AMS buffer. If needed, pH was adjusted to 7.0 with 1 M Tris/HCl pH 8.8 and the oxidation state of cysteine residues was assessed as described above for isolated mitochondria.

## Cycloheximide chase

Yeast strains were inoculated in YPD medium and grown overnight at 37 °C with shaking at 150 rpm. Overnight cultures were diluted in fresh medium and grown until early log phase (OD$_{600}$ = 0.4). Protein synthesis was inhibited by addition of 200 µg/mL cycloheximide, and cultures were further grown under the same conditions. At the indicated time points, 3 OD$_{600}$ cells were collected, washed with 1 mL sterile water and frozen at −80 °C. After the last sample was collected, total cell extracts were prepared and 0.3–0.6 OD$_{600}$ were loaded per lane. Analysis was performed via SDS-PAGE and immunodecoration.

## In vitro import

Tim17, Tim22 and Dbi1, cloned in pGEM4 vector under Sp6 promoter, were synthesized in vitro in the presence of

[35]S-methionine using TNT-coupled transcription/translation system (Promega).

Isolated mitochondria were resuspended in SI buffer containing 0.5 mg/mL BSA and energized with 2.5 mM ATP, 5 mM NADH, 10 mM CP and 0.1 mg/mL CK for 3 min at 25 °C before in vitro synthesized precursor protein was added. After the indicated time periods, aliquots were removed, and import was stopped by 1:10 dilution in ice-cold SH-buffer (20 mM HEPES-KOH, 0.6 M sorbitol, pH 7.2) containing 1 μM valinomycin. After the last time point, all samples were treated for 15 min on ice with PK. PK digestion was stopped by the addition of 1 mM PMSF. After 5 min on ice, mitochondria were reisolated by centrifugation (10 min, 18,000 × g, 4 °C), resuspended in AMS buffer, and incubated for 5 min at 95 °C. Where indicated, samples were supplemented with 15 mM mmPEG$_{24}$ and treated as described before for thiol modification. Samples were analyzed by SDS-PAGE and autoradiography. Dbi1 imports were analyzed on NuPAGE 4–12% Bis-Tris gradient gels with MES running buffer (Thermo Fisher).

## Antibody generation

Dbi1 antibody was raised against the C-terminal peptide (CNYKKLSNDGETQAN) coupled to keyhole limpet hemocyanin (KLH) by injection into rabbits. To validate the specificity of the obtained serum, WT and Δdbi1 mitochondria were resolved by SDS-PAGE, transferred onto nitrocellulose membrane and the resulting stripes were used for western blot. Dbi1-specific antibodies were affinity-purified from the serum using the affinity matrix made by coupling the Dbi1 peptide to SulfoLink beads.

The first 60 amino acid residues of Tim14 were fused to GST, the fusion protein was recombinantly expressed and purified, and injected into rabbits for the generation of Tim14-specific serum. Tim14-specific antibodies were affinity-purified before use.

## Miscellaneous

The experiments shown are typical representatives of results obtained in at least three independent experiments. To ensure reproducibility, the critical experiments were repeated by two researchers, in some cases even in two different laboratories. Yeast strain and tools generated here will be freely distributed to the members of the academic community.

# Data availability

This study includes no data deposited in external repositories.

The source data of this paper are collected in the following database record: biostudies:S-SCDT-10_1038-S44319-024-00349-6.

# Peer review information

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

## Acknowledgements

The authors thank the members of the Mokranjac laboratory and the Mito Club for support and discussions, Petra Robisch for expert technical assistance, Pavel Dolezal for help with the bioinformatic analysis, Jan Riemer for critical comments on the study, Nikolaus Pfanner and Johannes M Herrmann for sharing reagents and protocols, and Naiyareen Fareeza Mayeen for help with some experiments. This work was supported by Deutsche Forschungsgemeinschaft (project MO1944/3-1 to DM and project HE3462/4-1 to KH) and by Walter and Monika Neupert Foundation (a doctoral student fellowship to SB).

## Author contributions

**Soraya Badrie**: Data curation; Formal analysis; Funding acquisition; Validation; Investigation; Visualization; Methodology; Writing—review and editing. **Kai Hell**: Formal analysis; Supervision; Funding acquisition; Investigation; Writing—review and editing. **Dejana Mokranjac**: Conceptualization; Data curation; Formal analysis; Supervision; Funding acquisition; Validation; Investigation; Visualization; Methodology; Writing—original draft; Project administration; Writing—review and editing.

Source data underlying figure panels in this paper may have individual authorship assigned. Where available, figure panel/source data authorship is listed in the following database record: biostudies:S-SCDT-10_1038-S44319-024-00349-6.

## Funding

## Disclosure and competing interests statement

The authors declare no competing interests.

# Expanded View Figures

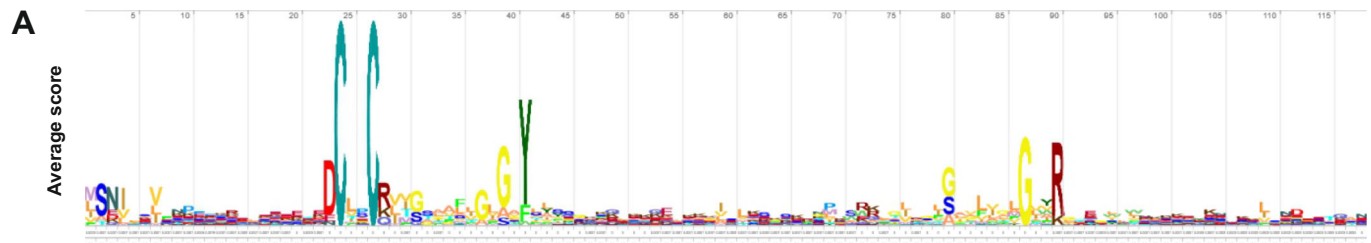

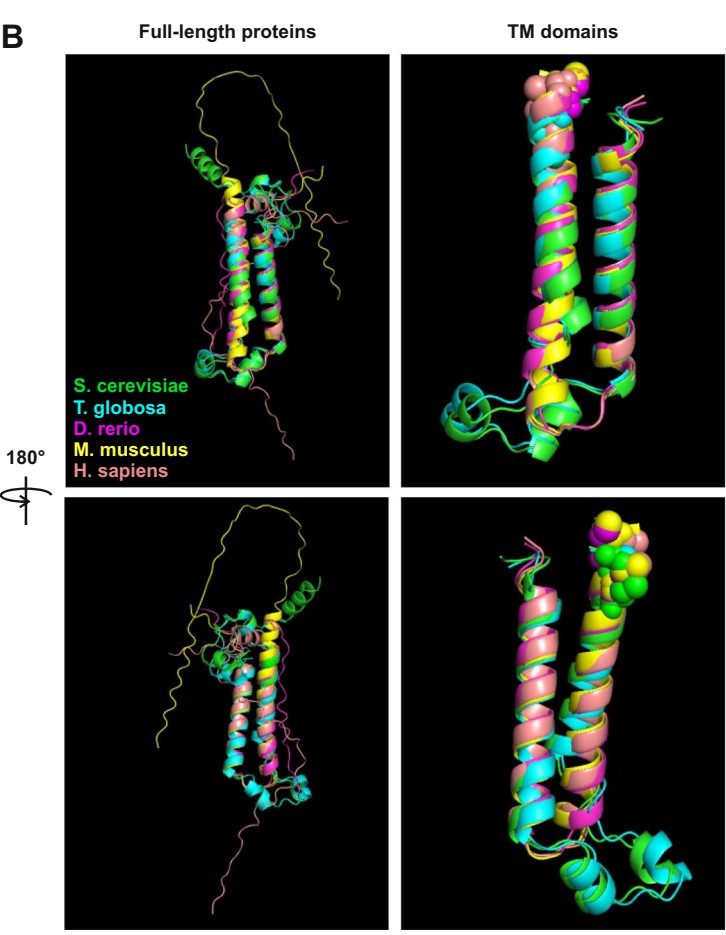

**Figure EV1. Evolutionary conservation of Dbi1.**

(A) *Saccharomyces cerevisiae* YDL157c protein sequence was used to search UniProtKB database using JACKHHMER online tool (https://www.ebi.ac.uk/Tools/hmmer/search/jackhmmer). After five iterations, 1136 proteins were identified, 1128 of which were eukaryotic. Screenshot of the obtained sequence conservation is shown. (B) AlphFold3 generated models of Dbi1 from *Saccharomyces cerevisiae* and its homologs from *Torulaspora globosa*, *Danio rerio*, *Mus musculus* and *Homo sapiens* were aligned using PyMOL. Cysteine residues are shown as spheres.

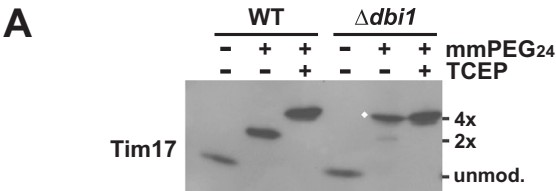

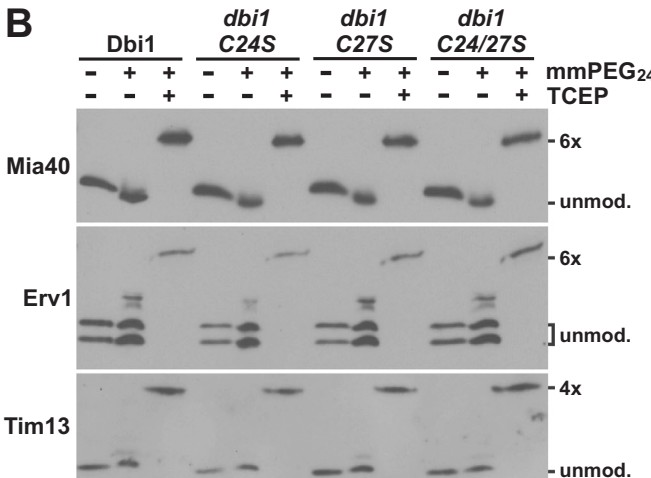

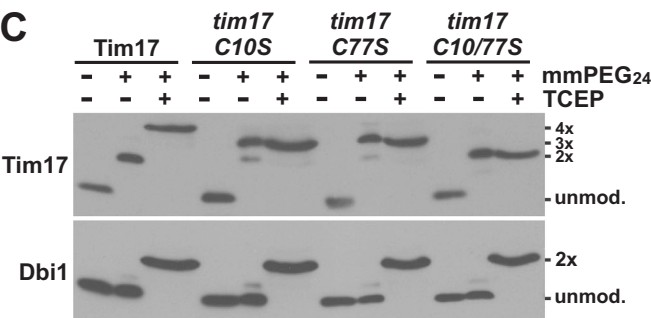

**Figure EV2. Thiol modification in Dbi1- and Tim17 cysteine mutants.**

(A) In vivo oxidation state of cysteine residues in cellular proteins was trapped by TCA-precipitation. Samples were solubilized in SDS-containing buffer and incubated with the free thiol reactive reagent methyl-polyethylene glycol-maleimide (mmPEG$_{24}$). One sample was fully reduced with TCEP at 96 °C prior to incubation with mmPEG$_{24}$. Samples were analyzed by SDS-PAGE and western blotting. White diamond highlights the fully reduced Tim17 in cells lacking Dbi1. (B) Isolated mitochondria containing indicated Dbi1 variants were solubilized in SDS-containing buffer and subsequently treated as in (A). (C) As in (B), with the difference that mitochondria containing Tim17 cysteine mutants were analyzed.

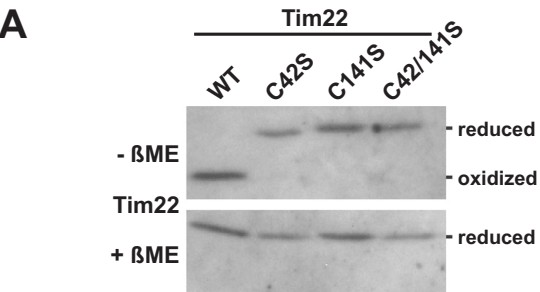

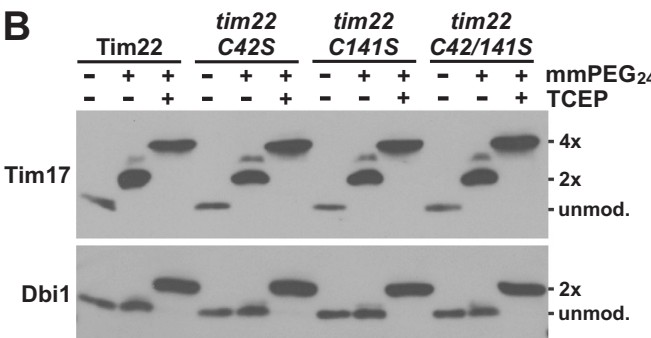

**Figure EV3. Analysis of Tim22 cysteine mutants.**

(A) Wild type and mitochondria containing indicated variants of Tim22 were dissolved in reducing (+ßME) or nonreducing (-ßME) Laemmli buffer and analyzed by SDS-PAGE and western blotting. (B) Isolated mitochondria containing indicated Tim22 variants were dissolved in SDS-containing buffer and incubated with methyl-polyethylene glycol-maleimide (mmPEG$_{24}$). One sample was fully reduced with TCEP at 96 °C prior to incubation with mmPEG$_{24}$. Samples were analyzed by SDS-PAGE and western blotting.

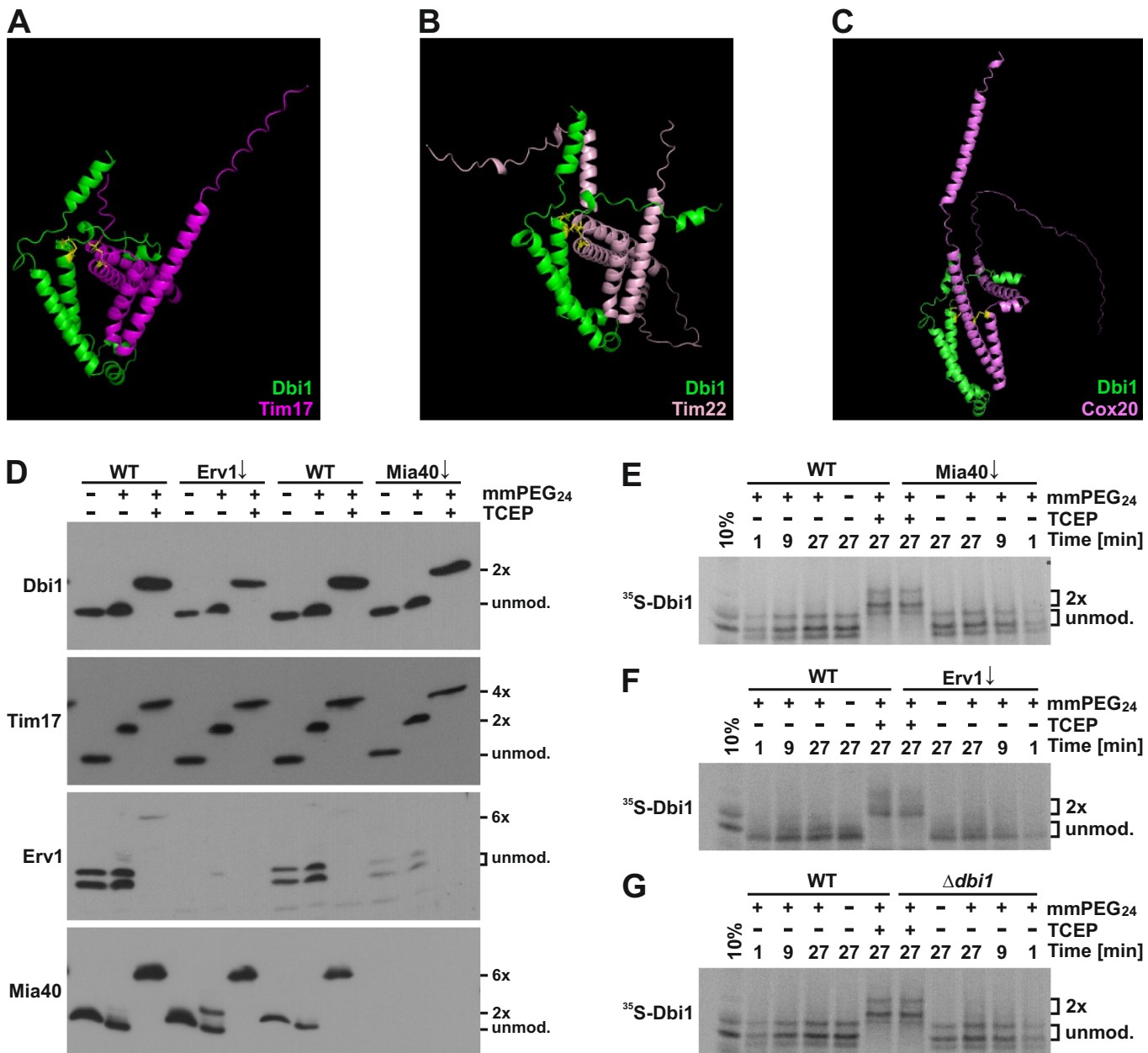

**Figure EV4.  Role of Dbi1 in oxidation and assembly of inner membrane proteins.**

AlphaFold3 generated models of Dbi1 in complex with Tim17 (**A**), Tim22 (**B**), and Cox20 (**C**). Cysteine residues are shown as yellow sticks. (**D**) Isolated mitochondria were solubilized in SDS-containing buffer and incubated with mmPEG$_{24}$, where indicated. One sample was fully reduced with TCEP at 96 °C prior to labeling with mmPEG$_{24}$. Samples were analyzed by SDS-PAGE and western blotting using indicated antibodies. (**E–G**) $^{35}$S-labeled Dbi1 was imported into isolated mitochondria. At indicated time points, samples were taken out, import was stopped and samples were treated with proteinase K to remove all nonimported material. Protease digestion was stopped by incubation with PMSF, mitochondria were reisolated, solubilized in SDS-containing buffer and incubated with mmPEG$_{24}$, where indicated. One sample was reduced with TCEP at 96 °C prior to incubation with mmPEG$_{24}$. Samples were analyzed by SDS-PAGE followed by autoradiography.

