## [Peer Review File · EMBO Reports]

Dbi1 is an oxidoreductase and an assembly chaperone for mitochondrial inner membrane proteins

Dejana Mokranjac, Kai Hell, and Soraya Badrie

Corresponding author(s): Dejana Mokranjac (mokranjac@bio.lmu.de)

Review Timeline:

Submission Date:	15th May 24
Editorial Decision:	20th Jun 24
Revision Received:	16th Sep 24
Editorial Decision:	31st Oct 24
Revision Received:	5th Nov 24
Accepted:	25th Nov 24

Editor: Deniz Senyilmaz Tiebe

Transaction Report:

Dear Dr. Mokranjac,

Thank you for transferring your research manuscript to our journal, which was now seen by three referees, whose reports are copied below.

Referees express interest in the proposed role of Dbi as an assembly chaperone for mitochondrial inner membrane. However, they also raise significant concerns that need to be addressed to consider publication here.

I find the reports informed and constructive, and believe that addressing the concerns raised will significantly strengthen the manuscript. As the reports are below, and I think all points need to be addressed, I will not detail them here.

Given these positive recommendations, we would like to invite you to submit a revised manuscript. Please revise your manuscript with the understanding that the referee concerns (as in their reports) must be fully addressed and their suggestions taken on board. Please address all referee concerns in a complete point-by-point response. Acceptance of the manuscript will depend on a positive outcome of a second round of review. It is EMBO reports policy to allow a single round of major experimental revision only and acceptance or rejection of the manuscript will therefore depend on the completeness of your responses included in the next, final version of the manuscript.

We realize that it is difficult to revise to a specific deadline. In the interest of protecting the conceptual advance provided by the work, we recommend a revision within 3 months. Please discuss the revision progress ahead of this time with me if you require more time to complete the revisions, or if you have questions or comments regarding the revision (also by video chat).

1. A data availability section providing access to data deposited in public databases is missing (where applicable).
2. Your manuscript contains statistics and error bars based on $n=2$. Please use scatter plots in these cases.

You can submit the revision either as a Scientific Report or as a Research Article. For Scientific Reports, the revised manuscript can contain up to 5 main figures and 5 Expanded View figures, and it should not exceed 27000 characters. If the revision leads to a manuscript with more than 5 main figures it will be published as a Research Article. In this case the Results and Discussion section should be separate. If a Scientific Report is submitted, these sections have to be combined. This will help to shorten the manuscript text by eliminating some redundancy that is inevitable when discussing the same experiments twice. In either case, all materials and methods should be included in the main manuscript file.

4) a .docx formatted letter INCLUDING the reviewers' reports and your detailed point-by-point responses to their comments. As part of the EMBO publication's Transparent Editorial Process, EMBO reports publishes online a Review Process File (RPF) to accompany accepted manuscripts. This File will be published in conjunction with your paper and will include the referee reports, your point-by-point response and all pertinent correspondence relating to the manuscript.

<https://www.embopress.org/page/journal/14693178/authorguide#transparentprocess>

5) a complete author checklist, which you can download from our author guidelines

<https://www.embopress.org/page/journal/14693178/authorguide>. Please insert information in the checklist that is also reflected in the manuscript. The completed author checklist will also be part of the RPF.

6) Please note that all corresponding authors are required to supply an ORCID ID for their name upon submission of a revised manuscript (<<https://orcid.org/>>). Please find instructions on how to link your ORCID ID to your account in our manuscript tracking system in our Author guidelines

<<https://www.embopress.org/page/journal/14693178/authorguide#authorshipguidelines>>

7) Before submitting your revision, primary datasets produced in this study need to be deposited in an appropriate public database (see <https://www.embopress.org/page/journal/14693178/authorguide#datadeposition>). Please remember to provide a reviewer password if the datasets are not yet public. The accession numbers and database should be listed in a formal "Data Availability" section placed after Materials & Method (see also

<https://www.embopress.org/page/journal/14693178/authorguide#datadeposition>). Please note that the Data Availability Section is restricted to new primary data that are part of this study. * Note - All links should resolve to a page where the data can be accessed. *

Additional information on source data and instruction on how to label the files are available:

<https://www.embopress.org/page/journal/14693178/authorguide#sourcedata>

9) Our journal encourages inclusion of *data citations in the reference list* to directly cite datasets that were re-used and obtained from public databases. Data citations in the article text are distinct from normal bibliographical citations and should directly link to the database records from which the data can be accessed. In the main text, data citations are formatted as follows: "Data ref: Smith et al, 2001" or "Data ref: NCBI Sequence Read Archive PRJNA342805, 2017". In the Reference list, data citations must be labeled with "[DATASET]". A data reference must provide the database name, accession number/identifiers and a resolvable link to the landing page from which the data can be accessed at the end of the reference. Further instructions are available at <http://www.embopress.org/page/journal/14693178/authorguide#referencesformat>

10) Regarding data quantification (see Figure Legends:

<https://www.embopress.org/page/journal/14693178/authorguide#figureformat>)

12) Please also note our reference format:

13) All Materials and Methods need to be described in the main text. We would encourage you to use 'Structured Methods', our new Methods format. According to this format, the Methods section should include a Reagents and Tools Table (listing key reagents, experimental models, software and relevant equipment and including their sources and relevant identifiers) followed by a Methods and Protocols section in which we encourage the authors to describe their methods using a step-by-step protocol format with bullet points, to facilitate the adoption of the methodologies across labs. More information on how to adhere to this format as well as downloadable templates (.doc or .xls) for the Reagents and Tools Table can be found in our author guidelines: <<https://www.embopress.org/page/journal/14693178/authorguide#manuscriptpreparation>>.

An example of a Method paper with Structured Methods can be found here: <https://www.embopress.org/doi/full/10.1038/s44320-024-00037-6#sec-4>

I look forward to seeing a revised version of your manuscript when it is ready. Please let me know if you have questions or comments regarding the revision.

Kind regards,

Deniz Senyilmaz Tiebe

Deniz Senyilmaz Tiebe, PhD
Scientific Editor
EMBO Reports

Referee #1:

The mitochondrial inner membrane contains two major membrane-integrated protein translocases that recognize and translocate numerous nuclear-encoded precursor proteins, the presequence translocase (TIM23 complex) and the carrier translocase (TIM22 complex). Two core components, Tim17 of the TIM23 complex and Tim22 of the TIM22 complex, contain disulfide bonds. However, these disulfide bonds are not inserted by the well-established MIA disulfide relay of mitochondria.

The authors of this study report the identification of a novel oxidoreductase of mitochondria, termed Dbi1. In a series of elegant experiments, they show that Dbi1 inserts the disulfide bonds into Tim17 and Tim22 and additionally functions in a chaperone-like manner to promote assembly of the translocases. This is a remarkable finding and reveals a new pathway for the oxidation and assembly of crucial inner membrane proteins.

This paper by experts of the field is of excellent quality with clearly structured and well-controlled experiments. The text is written in a convincing manner and accessible to a broad audience.

I have only minor points to add:

1. The name of Dbi1 is explained in the first chapter of the Results section. It would be helpful for the reader to have the name of the new protein explained in the Abstract and Introduction.

2. Legend of Figure 1F: 10, 20 and 50 microgram of isolated mitochondria (protein amount) were ..

I assume that the authors refer to the total protein amount of the mitochondria and thus this should be indicated, e.g. as shown above.

The same applies to the legend of Figure 2F.

Referee #2:

Summary

The manuscript by Badrie et al. reports about a mitochondrial inner membrane protein named Dbi1 with oxidoreductase activity which is involved in the biogenesis of two essential inner membrane proteins Tim17 (core subunit of the presequence translocase TIM23) and Tim22 (core subunit of the carrier translocase TIM22) containing two cysteine residues each, which are

oxidized to from an intramolecular disulfide bond. This work is highly significant and was not reported before. In addition to the oxidoreductase activity the authors propose that Dbi1 acts as membrane protein chaperone. In contrast to specific assembly factors, so far little is known about general membrane protein chaperones which are crucial for the assembly of multiple different complexes. Moreover, the Dbi1 seems to be involved in further processes (see previous publications and other posted manuscripts on Dbi1 below) and the proposed function as membrane protein chaperone could be a primary function. Therefore, this manuscript represents an extremely important contribution to the molecular biology community. The majority of the results are robustly documented using independent lines of experimental evidence.

Major points

1 Concerning the function (and name) of Dbi1 (disulfide bond formation in inner membrane proteins). Can the authors exclude that Dbi1 plays an additional role as protein disulfide-isomerase for soluble intermembrane space or even matrix proteins? Do the *dbi1C24S* and *dbi1C27S* mutants form mixed disulfide intermediates with substrates (the authors could analyze this e.g. by non-reducing SDS-PAGE)? Are there other mitochondrial membrane proteins with conserved cysteines which could be substrates of Dbi1? In case it would be feasible for this study, it would be interesting to analyze the substrate pool by trapping of mixed disulfide intermediates followed by identification by mass spectrometry. The authors should speculate on the substrate spectrum of Dbi1.

2 In the beginning of the results section the author write that Dbi1 is conserved, but do not specify the conservation. The authors should discuss this and present a proper analysis as Expanded View figure.

3 Does Dbi1 play a direct role for the biogenesis/oxidation of Tim22? How does the deletion of DBI1 affect the protein level of Tim22 respective the TIM22 complex (minus/plus CHX)? Can the authors isolate Dbi1-Tim22 intermediates similar to Dbi1-Tim17?

Regarding Fig. 5 and the manuscript:

4 First of all, what is the evidence that Tim17/Tim22 precursors are imported via Mia40? In case the authors would like to summarize the biogenesis pathway of Tim17/Tim22 precursors then small TIM chaperones and the TIM22 complex are clearly lacking.

5 Second of all, the authors should analyze how Dbi1 is oxidized (by Mia40, Erv1 and/or other intermembrane space proteins?) to enable the transfer of the disulfide bonds to its substrates.

6 Third of all (and most problematic), oxidized Dbi1 is depicted to react with reduced Tim17/Tim22 to form oxidized Dbi1 (likely the idea was to draw reduced Dbi1 in this case) and oxidized Tim17/Tim22.

7 In the discussion the authors reported that Dbi1 influences mitochondrial translation (Dickinson et al., 2022), but that they were not able to recapitulate the growth phenotype. Interestingly, YDL157C was previously already classified as *pet*-mutant with unspecific loss of mitochondrial DNA (Merz and Westermann, 2009 Genome Biology, doi:10.1186/gb-2009-10-9-r95). Recently strong growth defects for *ydl157c* deletion yeast strains in two different genomic backgrounds together with a proposed role of Ydl157c in mitochondrial copper metabolism and Cox2 biogenesis were posted (Soares et al., 2024 bioRxiv, doi: 10.1101/2023.06.12.544600). The authors should present their own data on the growth of the *ydl157c* deletion yeast strain(s) and discuss all the previous publications and the recent post related to Dbi1/Ydl157c.

8 Moreover, it was published that Ydl157c is localizing to the peroxisomal membrane or membrane periphery (Yifrach et al., 2022 Molecular Systems Biology, doi: 10.15252/msb.202211186). Therefore, it might be interesting to perform a subcellular fractionation to acquire a basic characterization of the potentially dual localized mitochondrial and peroxisomal protein. Of course, also this report should be also adequately discussed.

Minor points

1 Page 5, RESULTS, Dbi1 is a novel...: The authors write "the ... crosslink ...had a slightly slower running behavior". The reviewer considers this as lab slang.

2 Fig. 4B: It might help to indicate in the figure labelling, that most lanes were subjected to +mmPEG24 treatment.

3 Fig. 4F and H: One can predict that the mutants have a significant growth defect. It would be better to present plates where the strains expressing wildtype Tim17 and wildtype Tim22 form large single colonies, such that one can judge the growth defect of the cysteine mutants.

Referee #3:

Mitochondrial protein import into the intermembrane space is mainly mediated by the disulfide relay system composed of the oxidoreductase Mia40 and the sulfhydryl oxidase Erv1, which oxidize and thereby trap in particular small soluble proteins with characteristic CXC motifs in the IMS. Here, Badrie et al characterize a small mitochondrial protein that localizes in the inner

membrane and interacts with Tim17 and Tim22 during their biogenesis to introduce disulfide bonds close to the inner membrane lipid bilayer. The identification of an additional oxidoreductase activity in the IMS that plays a role in mitochondrial protein biogenesis is of general interest for the mitochondrial research field. However, at the moment there are some inconsistencies and open questions that need to be addressed to support the major claims of the current manuscript.

Major concerns:

- It is puzzling that the x-linking analysis detects Dbi1 with Tim23, but in the native pull-down Dbi1 is not interacting with Tim23, but only with Tim17. Is the interaction of Dbi1 and Tim23 detectable when the Tim23 enrichment is performed in the *tim23-87A5* mutant? What is special about this mutant so that Dbi1 is specifically interacting here with Tim23? Along the same lines: Does Dbi1 x-link to Tim23 in the Tim17 cysteine mutants?
- Deletion of Dbi1 results in approximately half of Tim17 being oxidized. The authors speculate in their discussion that this might be due to Erv1 that partially takes over when Dbi1 is missing. This can be very easily tested by using an additional deletion of Erv1 and this would strongly support the interplay between different oxidoreductases in the IMS.
- The shift of the x-link due to the additional his tag on Dbi1 is not really visible in figure 1B. Can the authors load the two relevant lanes on a separate gel next to each other, to make the comparison of the sizes easy and the difference well visible? Also the x-linking pattern of Tim23 in WT changes strongly between figure 1A/B and figure 1G. What is the reason for this?
- The Dbi1 signal is already decreasing significantly after addition of proteinase K to intact mitochondria (figure 1D). Could Dbi1 be also localized to the outer membrane? Are there PK fragments that can be used as a readout for protease accessibility?
- While the authors demonstrate the oxidoreductase function of Dbi1 convincingly, data on the additional assembly chaperone function is weak and mainly deduced from a publication on the human homologue DMAC1 (Stroud et al., 2016). The claim of this function in yeast is therefore not supported and should be only phrased as a hypothesis in the discussion. Also the title of the manuscript should be changed to reflect the provided data.
- The analysis of the double deletions in figure 4 nicely illustrates the strong defects. However, these growth analyses do not allow conclusions on cell viability and therefore the statements about cell death and lethality should be omitted.

Minor points:

- The authors observe an additional x-link with Tim23 that likely comes from a 14 kDa protein. They then state that the ORF now termed Dbi1 fulfills "all the requirements". What are the other requirements besides the fit in kDa?
- In the materials and methods the radiolabeled lysates are generated without the addition of DTT. These would result in non-import competent precursors (of substrates of the Mia40 import pathway). Did the authors add DTT or is Tim17 import-competent without addition of DTT?
- The authors make a quantitative statement about increased protein steady state levels without providing quantifications (Fig. 2F). Please provide the quantification or preferably refrain from such quantitative statements (the protein increase is clearly visible).
- Typo on page 7 Mi40 instead of Mia40 and page 9 Trim22 instead of Tim22.
- The authors state that they cannot reproduce the data from Dickinson et al. Can they speculate why this might be the case? Is it another strain background or maybe the false discovery rate of high-throughput data?

Dear Dr Senyilmaz Tiebe,

We thank you and the three anonymous Referees for the careful evaluation, constructive criticism and thoughtful comments on our manuscript. The suggestions and comments of the Referees certainly helped to clarify several points in the manuscript and significantly contributed to its improvement. Below we address point-by-point the various comments and suggestions raised by the Referees.

Referee #1:

The mitochondrial inner membrane contains two major membrane-integrated protein translocases that recognize and translocate numerous nuclear-encoded precursor proteins, the presequence translocase (TIM23 complex) and the carrier translocase (TIM22 complex). Two core components, Tim17 of the TIM23 complex and Tim22 of the TIM22 complex, contain disulfide bonds. However, these disulfide bonds are not inserted by the well-established MIA disulfide relay of mitochondria. The authors of this study report the identification of a novel oxidoreductase of mitochondria, termed Dbi1. In a series of elegant experiments, they show that Dbi1 inserts the disulfide bonds into Tim17 and Tim22 and additionally functions in a chaperone-like manner to promote assembly of the translocases. This is a remarkable finding and reveals a new pathway for the oxidation and assembly of crucial inner membrane proteins.

This paper by experts of the field is of excellent quality with clearly structured and well-controlled experiments. The text is written in a convincing manner and accessible to a broad audience.

We thank the Referee for his/her very positive evaluation of our work.

I have only minor points to add:

1. The name of Dbi1 is explained in the first chapter of the Results section. It would be helpful for the reader to have the name of the new protein explained in the Abstract and Introduction.

We thank the Referee for this comment – we introduced the name in the Introduction. In the Abstract, due to space limitations, we underlined the letters that form the Dbi acronym.

2. Legend of Figure 1F: 10, 20 and 50 microgram of isolated mitochondria (protein amount) were .. I assume that the authors refer to the total protein amount of the mitochondria and thus this should be indicated, e.g. as shown above.

The same applies to the legend of Figure 2F.

We apologize for the confusion. Indeed, protein amount of mitochondria was meant and we modified the text accordingly.

Referee #2:

Summary

The manuscript by Badrie et al. reports about a mitochondrial inner membrane protein named Dbi1 with oxidoreductase activity which is involved in the biogenesis of two essential inner membrane proteins Tim17 (core subunit of the presequence translocase TIM23) and Tim22 (core subunit of the carrier translocase TIM22) containing two cysteine residues each, which are oxidized to form an intramolecular disulfide bond. This work is highly significant and was not reported before. In addition to the oxidoreductase activity the authors propose that Dbi1 acts as membrane protein chaperone. In contrast to specific assembly factors, so far little is known about general membrane protein chaperones which are crucial for the assembly of multiple different complexes. Moreover, the Dbi1 seems to be involved in further processes (see previous publications and other posted manuscripts on Dbi1 below) and the proposed function as membrane protein chaperone could be a primary function. Therefore, this manuscript represents an extremely important contribution to the molecular biology community. The majority of the results are robustly documented using independent lines of experimental evidence.

We thank the Referee for his/her very positive evaluation of our work.

Major points

1 Concerning the function (and name) of Dbi1 (disulfide bond formation in inner membrane proteins). Can the authors exclude that Dbi1 plays an additional role as protein disulfide-isomerase for soluble intermembrane space or even matrix proteins?

The data presented in this study demonstrate the role of Dbi1 as an oxidoreductase involved in disulfide bond formation of at least two inner membrane proteins – Tim17 and Tim22. In response to this Referee's comment below, we identified Cox20 as an additional substrate of Dbi1, see below. In contrast, we saw no effect of Dbi1 on the disulfide relay system or on its substrates. We named the protein based on these findings. With that said, it is rarely, if ever, possible to completely exclude additional functions of proteins. However, addressing them would, in our opinion, go beyond the scope of this manuscript, which aimed at the identification and initial characterization of the protein.

Do the dbi1C24S and dbi1C27S mutants form mixed disulfide intermediates with substrates (the authors could analyze this e.g. by non-reducing SDS-PAGE)?

We agree with the Referee that this is a very important point. Following Referee's suggestion, we solubilized mitochondria isolated from WT, *dbi1C24S*, *dbi1C27S* and *dbi1C24S/C27S* cells in nonreducing (-βME) and reducing (+βME) Laemmli buffers and analyzed the respective lysates by SDS-PAGE followed by western blot using antibodies against Tim17 and Tim22, Figure 1 for the Referees. The oxidized and reduced forms of Tim22 can easily be distinguished in this assay and we indeed see that ca. half of Tim22 is found in the reduced state in all *dbi1* mutants, as we already showed in Fig. 4D of the initially submitted manuscript. We were, however, unable to detect any specific signal in the higher molecular weight range which could represent a mixed disulfide intermediate between Dbi1 and Tim22. We were similarly unable to detect a mixed disulfide intermediate between Dbi1 and Tim17. It is possible, even likely, that these intermediates are very short-lived and therefore hard to detect.

Figure 1 for the Referees. Isolated mitochondria were solubilized in non-reducing (-βME) and reducing (+βME) buffers and the lysates analyzed by SDS-PAGE and western blotting using antibodies against Tim22 (upper panel) and Tim17 (lower panel).

Are there other mitochondrial membrane proteins with conserved cysteines which could be substrates of Dbi1? In case it would be feasible for this study, it would be interesting to analyze the substrate pool by trapping of mixed disulfide intermediates followed by identification by mass spectrometry. The authors should speculate on the substrate spectrum of Dbi1.

This is another very important point. After we have seen the effect of Dbi1 on Tim17, we wondered whether this effect is limited to Tim17 and reasoned that a structurally related protein Tim22 could be another substrate of Dbi1, which turned out to be true, as we presented in the manuscript. As mentioned above, we were unable to detect mixed disulfide intermediates between Dbi1 and either Tim22 or Tim17. We, however, anyway attempted to address the Referee's question using mass spectrometry after a regular pull down. Unfortunately, we were not successful using this approach. One of the possible reasons is the very hydrophobic nature of both Dbi1 and its so far identified substrates. We, thus, chose to directly investigate several potential candidates for which we had antibodies available. Using labelling of cysteine residues with mmPEG₂₄ in WT and Δ *dbi1* mitochondria, we indeed found part of Cox20 in the reduced state in the absence of Dbi1, whereas the protein was fully oxidized in WT mitochondria. In contrast, we saw no influence of Dbi1 on the oxidation states of Rip1 or Qcr6. This suggests that the spectrum of Dbi1 substrates is not limited to Tim17 and Tim22 and further supports the notion that Dbi1 substrates are inner membrane proteins with disulfide bonds in close vicinity of the inner membrane. These results are now presented in the manuscript text and shown as new Figure 5A.

2 In the beginning of the results section the author write that Dbi1 is conserved, but do not specify the conservation. The authors should discuss this and present a proper analysis as Expanded View figure. We thank the Referee for this comment – we identified Dbi1 homologues using JACKHMMER. Starting with the yeast Dbi1 protein sequence, we searched UniProtKB database using JACKHMMER online tool (<https://www.ebi.ac.uk/Tools/hmmer/search/jackhmmer>). After five iterations, we identified 1136 proteins, 1128 of which were eukaryotic. Sequence conservation is now included in new Figure EV1A. We also used AlphFold3 to create models of yeast Dbi1 and several of its homologues, new Figure EV1B, which demonstrate a remarkable structural conservation of the two transmembrane segments as well as of the position of the CxxC motif.

3 Does Dbi1 play a direct role for the biogenesis/oxidation of Tim22? How does the deletion of DBI1 affect the protein level of Tim22 respective the TIM22 complex (minus/plus CHX)? Can the authors isolate Dbi1-Tim22 intermediates similar to Dbi1-Tim17?

We showed in Figure 4B that newly imported Tim22 remains in the reduced state in $\Delta dbi1$ mitochondria whereas it is fully oxidized after import into WT mitochondria. Furthermore, we found a part of the endogenous Tim22 in a reduced state both in $\Delta dbi1$ mitochondria as well as in *dbi1* CxxC motif mutants, Figures 4C and 4D. This, in our opinion, strongly supports a direct role of Dbi1 in biogenesis/oxidation

of Tim22. In contrast, *tim17* cysteine mutants for example had no influence on the oxidation state of Tim22, Figure 2 for the Referees.

Following Referee's suggestion, we attempted to demonstrate an interaction between Dbi1 and Tim22 in a coimmunoprecipitation experiment. To this end, we first solubilized WT mitochondria with digitonin and used antibodies against Dbi1 for coimmunoprecipitation and antibodies against Tim22, a generous gift from the laboratory of Dr. Nikolaus Pfanner (University of Freiburg), for the subsequent western blot. Unfortunately, we were unable to detect Tim22 in the bound fraction, due to the very strong background, Figure 3 for the Referees, left panel. We then generated a yeast strain expressing

an HA-tagged version of Tim22, isolated mitochondria and repeated the coimmunoprecipitation experiment now using antibodies against the HA tag for the subsequent western blot. However, also in this case, we were unable to detect Tim22 in the bound fraction, Figure 3 for the Referees, right panel. We are of the opinion that the detection of likely a very minor amount of Tim22 bound to Dbi1 is a technical issue. The levels of the TIM22 complex are several times lower than those of the TIM23 complex (Morgenstern et al, Cell Rep, 2017) which makes such an analysis harder for Tim22 compared to Tim17. Furthermore, we have affinity-purified antibodies against Tim17, which considerably improve such analyses. During the time

Figure 3 for the Referees. Mitochondria, isolated from WT, left panel, and cells expressing an HA-tagged version of Tim22, right panel, were solubilized in digitonin containing buffer and clarified lysates were incubated with affinity purified antibodies against Dbi1 prebound to ProteinA-Sepharose beads. Antibodies from the preimmune serum (PI) served as a negative control. After three washing steps, specifically bound proteins were eluted with reducing Laemmli buffer. Total, nonbound (NB) and bound (B) samples were analyzed by SDS-PAGE AND western blot using indicated antibodies. Total and NB samples contain 20% of the material present in B.

available for the revision of this manuscript, it was not possible to generate new Tim22 antibodies and affinity purify them.

Using the currently available tools, we were also unable to detect Tim22 in whole cell extracts and we, therefore, did not attempt to directly analyze the stability of Tim22 in WT and *Δdbi1* cells. However, we expect Tim22 to behave similarly to Tim17 in *Δdbi1* cells, especially in the light of the published observation that Tim22 gets destabilized in the absence of the disulfide bond (Okamoto et al, JBC, 2014).

Regarding Fig. 5 and the manuscript:

4 First of all, what is the evidence that Tim17/Tim22 precursors are imported via Mia40? In case the authors would like to summarize the biogenesis pathway of Tim17/Tim22 precursors then small TIM chaperones and the TIM22 complex are clearly lacking.

We agree with the Referee that the model presented in Figure 5 did not include all the steps of the biogenesis pathways of Tim17/Tim22 and that the TIM22 complex and the small TIM chaperones are clearly missing. To simplify the model, we now removed all the steps before insertion into the inner membrane and describe only the steps we believe Dbi1 is involved in.

Involvement of Mia40 in import of Tim17/Tim22 precursors was reported by Chacinska and Herrmann groups (Wrobel et al, MBoC, 2013; Ramesh et al, JCB, 2016; Wrobel et al, Sci Rep, 2016).

5 Second of all, the authors should analyze how Dbi1 is oxidized (by Mia40, Erv1 and/or other intermembrane space proteins?) to enable the transfer of the disulfide bonds to its substrates.

We agree with the Referee that this is a very important point. To this end, we analyzed the oxidation state of endogenous Dbi1 in mitochondria depleted of either Mia40 or of Erv1. Since both proteins are essential for viability of yeast cells, we used *GAL-MIA40* and *GAL-ERV1* yeast strains in which expression of Mia40 and Erv1, respectively, is under the control of the regulatable *GAL* promoter. Mia40 and Erv1 were depleted by growing cells in the absence of galactose (Terziyska et al, FEBS Lett, 2005; Mesecke et al, Cell, 2005). Depletion of either of the two proteins had no influence on the oxidation state of endogenous Dbi1 (and of Tim17). In contrast, the CPC motif of Mia40 was partly reduced in mitochondria depleted of Erv1, in agreement with the role of Erv1 in reoxidation of Mia40, and Erv1 levels were reduced in mitochondria depleted of Mia40, in agreement with the role of Mia40 in import of Erv1. This data is now included as Figure EV 4D.

We also analyzed oxidation of newly imported Dbi1. However, also in this setup, we saw no effect of Mia40, Erv1 or of Dbi1 itself on the oxidation of Dbi1 – newly imported Dbi1 was oxidized in mitochondria lacking Mia40, Erv1 or Dbi1 in the same way as in the corresponding WT mitochondria. We now included this data as Figure EV 4E-G.

Based on these findings, we conclude that none of the currently known protein oxidases in the IMS seem to be involved in oxidation of Dbi1. Therefore, how Dbi1 gets its disulfide bond remains currently unclear and it is certainly one of the most pressing questions to be addressed in the future. We hope that the Referee will agree with us that no pathway was ever completely clarified in one study.

6 Third of all (and most problematic), oxidized Dbi1 is depicted to react with reduced Tim17/Tim22 to form oxidized Dbi1 (likely the idea was to draw reduced Dbi1 in this case) and oxidized Tim17/Tim22. Even though this may look like an obvious mistake on our side, it was actually our intention to draw oxidized Dbi1 in this case. Namely, in mitochondria depleted of Tim23 and in *tim23G112L* mutant mitochondria, in which we saw increased interaction between Dbi1 and Tim17 (Figure 2C and 2D), we also saw that both Dbi1 and Tim17 were fully oxidized (Figure 3H and 3I). We concluded from these results that the reoxidation of Dbi1 apparently can take place while Dbi1 is still bound to its substrates. We modified the model to clarify this point and also discuss it in the text.

7 In the discussion the authors reported that Dbi1 influences mitochondrial translation (Dickinson et al., 2022), but that they were not able to recapitulate the growth phenotype. Interestingly, YDL157C was previously already classified as *pet*-mutant with unspecific loss of mitochondrial DNA (Merz and Westermann, 2009 Genome Biology, doi:10.1186/gb-2009-10-9-r95). Recently strong growth defects for *ydl157c* deletion yeast strains in two different genomic backgrounds together with a proposed role of Ydl157c in mitochondrial copper metabolism and Cox2 biogenesis were posted (Soares et al., 2024 bioRxiv, doi: 10.1101/2023.06.12.544600). The authors should present their own data on the growth

of the *ydl157c* deletion yeast strain(s) and discuss all the previous publications and the recent post related to *Dbi1/Ydl157c*.

We typically use YPH499 as the WT strain. We presented growth of *Δdbi1* cells in the YPH499 background on both fermentable and nonfermentable media at 30°C in Figures 4C and 4G. When we initially made *Δdbi1* cells in YPH499 background, we selected two independent clones in which *YDL157c* was deleted with a *HIS3* marker and two in which *YDL157c* was deleted with a *KAN* marker. All four clones behaved identically. Following the Referees suggestion, we now performed a more detailed growth analysis of *Δdbi1* cells. We now not only included the four independent clones in YPH499 background but also one in W303 background and one in BY4741 background. The latter two, along with the corresponding WT strains, were generously provided to us by the group of Dr. Christof Osman (LMU Munich). No obvious growth defect was observed for any of the six different deletion strains in three different WT backgrounds on either fermentable (YPD) or two different nonfermentable (YPG and YPLac) media at 24°C, 30°C or 37°C, Figure 4A for the

Figure 4 for the Referees. (A) Serial 10-fold dilutions of indicated yeast strains in logarithmic growth phase were prepared and spotted on plates with fermentable (glucose, YPD) and two different non-fermentable carbon sources (glycerol, YPG and lactate, YPLac). Plates were incubated at 24°C, 30°C or 37°C for 2-3 days. (B) Whole cells extracts of indicated cells were prepared and analyzed by SDS-PAGE followed by western blot using indicated antibodies.

Referees. Western blot analysis after separation of the corresponding whole cell extracts by SDS-PAGE confirmed that *Dbi1* protein was absent in all deletion strains, Figure 4B for the Referees. We would leave to the Editor the decision on whether to include this data as EV Figure into the manuscript.

We also established the protocol to label cysteine residues with mmPEG₂₄ in cell lysates, without the need for previous isolation of mitochondria. Using this approach, we confirmed that Tim17 is present in the reduced state in the absence of Dbi1 in all different backgrounds, Figure 5 for the Referees. Interestingly, in

whole cells, we observed that not only approximately half of Tim17 but rather its majority is present in the reduced state in Δ*dbi1* cells. We elaborate more on this point below, in the response to the second comment of Referee #3.

Therefore our statement that we could not reproduce the previously reported growth defects of Δ*dbi1* cells remains true. The lack of any obvious growth defect of Δ*dbi1* cells is in agreement with the absence of obvious growth defects observed in *tim17* and *tim22* cysteine mutants under typical laboratory conditions, Figure 6 for the Referees (and also Okamoto et al, JBC, 2014).

We apologize for not including the manuscript by Soares et al posted on the bioRxiv in the initial version of our

manuscript – it was posted in the very late stages of preparation of our manuscript and we missed to include it. This mistake on our side is now rectified.

8 Moreover, it was published that Ydl157c is localizing to the peroxisomal membrane or membrane periphery (Yifrach et al., 2022 Molecular Systems Biology, doi: 10.15252/msb.202211186). Therefore, it might be interesting to perform a subcellular fractionation to acquire a basic characterization of the potentially dual localized mitochondrial and peroxisomal protein. Of course, also this report should be also adequately discussed.

There are indeed several proteins which show clear dual localizations in both mitochondria and peroxisomes and whose functions in both organelles were carefully addressed and clarified. Fis1 is probably one prime example of such a protein. The first global analysis of protein localization in

budding yeast using chromosomal C-terminal GFP-tagging placed YDL157c in mitochondria (Huh et al, Nature, 2003). The same conclusion was reached by Dubreuil et al who used several different fluorescent protein tags on both N and C termini (Dubreuil et al, NAR, 2019). We can obviously not exclude a possibility that, under rather specific growth conditions used by Yifrach et al, a tagged version of Dbi1 may localize to peroxisomes. We would, however, like to note that the mentioned study did not include any mitochondrial marker during imaging and that it therefore remained unclear how mitochondria look under the same conditions. In addition, YDL157c has repeatedly been identified in highly purified mitochondria in proteomic studies (Sickman et al, PNAS, 2003; Reinders et al, 2006). The same is true for its human and mouse homologues (Rath et al, NAR, 2020). We also show here that Dbi1 localizes to mitochondrial inner membrane, the localization that is in full agreement with its here identified role in biogenesis of inner membrane proteins. These are ample evidence to support both the mitochondrial localization and function of Dbi1. We hope that the Referee will agree with us that addressing an additional and, in our opinion, still rather hypothetical peroxisomal localization of Dbi1 would go beyond the scope of this manuscript.

Minor points

1 Page 5, RESULTS, Dbi1 is a novel...: The authors write "the ... crosslink ...had a slightly slower running behavior". The reviewer considers this as lab slang.

We thank the Referee for pointing this out. We rephrased the sentence to now read "the uncharacterized crosslink was present but migrated slower, consistent with the size shift introduced by the His-tag".

2 Fig. 4B: It might help to indicate in the figure labelling, that most lanes were subjected to +mmPEG24 treatment.

We thank the Referee for noticing this. We relabelled the Figure to avoid confusion.

3 Fig. 4F and H: One can predict that the mutants have a significant growth defect. It would be better to present plates where the strains expressing wildtype Tim17 and wildtype Tim22 form large single colonies, such that one can judge the growth defect of the cysteine mutants.

We thank the Referee for pointing this out. We now show the plates that were incubated longer so that the differences in growth are more easily observed.

Referee #3:

Mitochondrial protein import into the intermembrane space is mainly mediated by the disulfide relay system composed of the oxidoreductase Mia40 and the sulfhydryl oxidase Erv1, which oxidize and thereby trap in particular small soluble proteins with characteristic CXC motifs in the IMS. Here, Badrie et al characterize a small mitochondrial protein that localizes in the inner membrane and interacts with Tim17 and Tim22 during their biogenesis to introduce disulfide bonds close to the inner membrane lipid bilayer. The identification of an additional oxidoreductase activity in the IMS that plays a role in mitochondrial protein biogenesis is of general interest for the mitochondrial research field. However, at the moment there are some inconsistencies and open questions that need to be addressed to support the major claims of the current manuscript.

We thank the Referee for his/her positive evaluation and hope that we were able to clarify the inconsistencies and open questions in the following sections.

Major concerns:

- It is puzzling that the x-linking analysis detects Dbi1 with Tim23, but in the native pull-down Dbi1 is not interacting with Tim23, but only with Tim17.

We fully agree with the Referee and we were equally puzzled by the initial observation that Dbi1 crosslinks to Tim23 but it only interacts with Tim17 in native pulldowns. This study was initiated with the idea to identify a novel, previously overlooked component of the TIM23 complex, however, the obtained results led us to a completely different direction. This is, in our opinion, one of the most exciting things about curiosity-driven research - one never knows where the experiments will lead you.

Is the interaction of Dbi1 and Tim23 detectable when the Tim23 enrichment is performed in the tim23-87A5 mutant?

Following the Referee's suggestion, we did NiNTA pulldowns from digitonin-solubilized mitochondria containing either wt or His-tagged version of Dbi1 in both Tim23 wt and tim23-87A5 mutant backgrounds, Figure 7 for the Referees. However, also under these conditions, we only see specific binding of Tim17 to the NiNTA beads. Neither wt

Tim23 nor Tim23-87A5 mutant was found in the bound fractions.

What is special about this mutant so that Dbi1 is specifically interacting here with Tim23?

We are not certain what, if anything, is special about this mutant, except that this is so far the only *tim23* mutant in which we observed this additional crosslink now identified as the Tim23-Dbi1 crosslink. We used AlphaFold3 to predict the structure of the putative Tim23-Tim17-Dbi1 complex and the model obtained would suggest that binding of Tim23 and Dbi1 to Tim17 are not mutually exclusive, Figure 8 for the Referees. The model further suggests that Arg91 of Tim23 forms a salt bridge with Asp8 in Tim17. Since Arg91 is the last amino acid residue mutated in *tim23-87A5* mutant, this salt bridge cannot be formed. This may alter the Tim17-Tim23 interaction sufficiently enough that the putative transfer of Tim17 from Dbi1 to Tim23 cannot be completed as efficiently as in WT. Furthermore, it appears that this intermediate is stable long enough that we could trap it using chemical

Figure 8 for the Referees. Structural model of the putative Tim23-Tim17-Dbi1 complex was generated by AlphaFold3. Tim23 is shown in cyan, Tim17 in magenta and Dbi1 in green. Residues mutated in *tim2387A5* cells are shown in orange and disulfide-bonded cysteine residues of Tim17 and Dbi1 in yellow.

The AlphaFold3 generated model also suggests that several lysine residues in Tim23 and Dbi1 are within the crosslinking distance of DSG (Tim23Lys143-Dbi1Lys52; Tim23Lys131-Dbi1Lys49), potentially explaining the observed crosslink.

Figure for referee with unpublished data and its description has been removed upon request by the authors.

- Deletion of Dbi1 results in approximately half of Tim17 being oxidized. The authors speculate in their discussion that this might be due to Erv1 that partially takes over when Dbi1 is missing. This can be very easily tested by using an additional deletion of Erv1 and this would strongly support the interplay between different oxidoreductases in the IMS.

We thank the Referee for this excellent suggestion. We could not perform exactly the suggested experiment as *ERV1* is an essential gene in yeast and therefore its deletion is lethal. However, we deleted *DBI1* in two temperature sensitive mutants of *erv1*, *erv1-1* and *erv1-2*, we generously obtained from the group of Dr.

Nikolaus Pfanner, and analyzed the oxidation state of Tim17, Figure 10 for the Referees. Since these mutants need to be incubated at the nonpermissive temperatures for 7h to induce the phenotype (Rissler et al, JMB, 2005), to simplify the subsequent analyses, we modified the assay to monitor the oxidation state of Tim17 using mmPEG₂₄ so that it can be used in whole cells, rather than in isolated mitochondria. When we analyzed the oxidation state of Tim17 in cells lacking Dbi1, we observed that the majority of Tim17 was present in the reduced state, rather than only approximately half of it as seen in isolated mitochondria. This suggests that re-oxidation can take place during the 3h needed to isolate mitochondria. Similar observation was made by Riemer and colleagues who showed that, while the entire pool of Mia40 is in an oxidized state in isolated mitochondria, a fraction of it is present in the reduced state *in vivo* (Kojer et al, EMBO J, 2012). When we analyzed *erv1ts* mutants, we saw very little reduced Tim17. Still, in WT essentially no reduced Tim17 was observed. These data suggest that Dbi1 has the major role in oxidation of Tim17 and that Erv1 may only contribute to a minor extent. In the light of these data, we removed the speculation on the interplay between different oxidoreductases in the IMS from the text.

- The shift of the x-link due to the additional his tag on Dbi1 is not really visible in figure 1B. Can the authors load the two relevant lanes on a separate gel next to each other, to make the comparison of the sizes easy and the difference well visible?

We agree with the Referee that the size shift is not very obvious in the blot shown in Figure 1B. We therefore show here another experiment in which, we think, the shift is better visible, Figure 11 for the Referees. We would also like to add that Figure 1B clearly shows that only the crosslink in tagged Dbi1 strain is specifically retained on the NiNTA beads, unambiguously demonstrating its identity as the crosslink between tagged Dbi1 (due to the tag) and Tim23-87A5 mutant protein (due to the antibody used for western).

Also the x-linking pattern of Tim23 in WT changes strongly between figure 1A/B and figure 1G. What is the reason for this?

The Referee probably refers to the strong Tim23 dimer visible in Figures 1A/B which is very weak in WT mitochondria in Figure 1G. The reason for this is that Figures 1A and B show crosslinking of Tim23 that is expressed from the plasmid, whereas, in Figure 1G, Tim23 is expressed from its chromosomal locus. Since *TIM23* is an essential gene, we make all the mutants using a plasmid shuffling strategy, as explained in Material and Methods. Even though we use centromeric, i.e. single copy, plasmids and express Tim23 from its endogenous promoter, the levels of Tim23 are always somewhat higher than upon expression from the chromosomal locus. This surplus of Tim23 makes homodimers, which can be easily detected by crosslinking. When Tim23 is expressed from its chromosomal locus, these dimers are barely visible, unless the conformation/composition of the TIM23 complex is affected, like in *Δdbi1* mitochondria.

- The Dbi1 signal is already decreasing significantly after addition of proteinase K to intact mitochondria (figure 1D). Could Dbi1 be also localized to the outer membrane?

We do agree with the Referee that the signal of Dbi1 decreased upon addition of PK to intact mitochondria. However, there is a similar decrease also of the Tim23 signal. Furthermore, the Tim23 fragment that is only observed after opening of the outer membrane, can also already be seen in the same sample. Therefore, we do not consider this as an argument for Dbi1 localization in the outer membrane but rather as a sign that a fraction of mitochondria already had a broken outer membrane. This is unfortunately a common phenomenon and it is essentially impossible to isolate all mitochondria with completely intact outer membrane. For this reason, the marker proteins are always analyzed in parallel.

Are there PK fragments that can be used as a readout for protease accessibility?

The antibodies we generated against Dbi1 recognize the C-terminal peptide of the protein and therefore, unfortunately, no stable fragment is to be expected. However, to address this point further, we synthesized Dbi1 *in vitro* in the presence of ³⁵S-methionine and performed submitochondrial localization after its import into isolated mitochondria. In this case, we see a clear, PK-stable fragment, only in mitoplasts and it is gone upon solubilization of both mitochondrial membranes. This result is now included in the manuscript as Figure 1E.

- While the authors demonstrate the oxidoreductase function of Dbi1 convincingly, data on the additional assembly chaperone function is weak and mainly deduced from a publication on the human homologue DMAC1 (Stroud et al., 2016). The claim of this function in yeast is therefore not supported and should be only phrased as a hypothesis in the discussion. Also the title of the manuscript should be changed to reflect the provided data.

We unfortunately have to disagree with the Referee on this point. Our statement on the chaperone function of Dbi1 is based on our findings a) that Dbi1 binds to the non-assembled pool of Tim17, b) that the interaction between Dbi1 and Tim17 is increased when Tim17 cannot assemble with Tim23 and c) that the levels of Dbi1 increase when Tim17 cannot assemble with Tim23. These are the *bona fide* criteria to classify a protein as an (assembly) chaperone.

- The analysis of the double deletions in figure 4 nicely illustrates the strong defects. However, these growth analyses do not allow conclusions on cell viability and therefore the statements about cell death and lethality should be omitted.

When the Tim17 and Tim22 shuffling strains are transformed with empty plasmids and the obtained transformants subsequently plated on media containing 5FOA, no viable colonies are obtained. This is in the literature commonly used as the criterion to say that yeast strains lacking functional copies of essential genes are dead and that deletions of essential genes are lethal. We, however, modified the statements to avoid use of terms "cell death" and "lethality".

Minor points:

- The authors observe an additional x-link with Tim23 that likely comes from a 14 kDa protein. They then state that the ORF now termed Dbi1 fulfills "all the requirements". What are the other requirements besides the fit in kDa?

The other requirements were that a candidate protein is present in mitochondria and that it is evolutionary conserved. YDL157c additionally drew our attention as, according to the CellMap, it has a genetic profile very similar to a number of TOM and TIM23 subunits.

- In the materials and methods the radiolabeled lysates are generated without the addition of DTT. These would result in non-import competent precursors (of substrates of the Mia40 import pathway). Did the authors add DTT or is Tim17 import-competent without addition of DTT?

Rabbit reticulocyte lysate we purchase from Promega contains 2mM DTT and we did not add any additional DTT.

- The authors make a quantitative statement about increased protein steady state levels without providing quantifications (Fig. 2F). Please provide the quantification or preferably refrain from such quantitative statements (the protein increase is clearly visible).

We removed “approximately 3-fold” from the sentence.

- Typo on page 7 Mi40 instead of Mia40 and page 9 Trim22 instead of Tim22.

We thank the Referee for noticing the typos, which we now corrected.

- The authors state that they cannot reproduce the data from Dickinson et al. Can they speculate why this might be the case? Is it another strain background or maybe the false discovery rate of high-throughput data?

We indeed used a different genetic background, which we thought could be the reason for the observed differences. However, in response to a similar comment of Referee #2, see above, we now also analyzed $\Delta dbi1$ cells generated in the same background, as previously described, and, in our hands, these cells behaved the same as our initially made deletion strains. We therefore currently cannot explain the observed growth differences.

We thank the Referees again for their insightful comments. We hope that we have addressed all their questions, comments and concerns in a satisfactory manner and that they will now find our manuscript suitable for publication.

Dear Dr. Mokranjac,

Thank you for submitting your revised manuscript. It has now been seen by two of the original referees.

As you can see, the referees find that the study is significantly improved during revision and recommend publication. However, I need you to address the points below before I can accept the manuscript.

- We believe that the format of the manuscript is better suited for our Reports format rather than Scientific Article, which needs to be updated accordingly during the resubmission.
- Please rename the 'Declaration of Interests' as 'Disclosure Statement and Competing Interests'.
- As per our format requirements, in the reference list, citations should be listed in alphabetical order and then chronologically, with the authors' surnames and initials inverted; where there are more than 10 authors on a paper, 10 will be listed, followed by 'et al.'. Please see <https://www.embopress.org/page/journal/14693178/authorguide#referencesformat>
- Please remove the Reagents and Tools table from the manuscript and upload it as a separate (word) document.
- The manuscript sections should be in the following order: Title page - Abstract & Keywords - Introduction - Results - Discussion - Methods - Data Availability - Acknowledgments - Disclosure Statement & Competing Interests - References - Figure Legends - (Main Tables with legends if applicable) - Expanded View Figure Legends.
- Papers published in EMBO Reports include a 'synopsis' and 'bullet points' to further enhance discoverability. Both are displayed on the html version of the paper and are freely accessible to all readers. The synopsis includes a short standfirst summarizing the study in 1 or 2 sentences (max 35 words) that summarize the paper and are provided by the authors and streamlined by the handling editor. I would therefore ask you to include your synopsis blurb and 3-5 bullet points listing the key experimental findings.
- In addition, please provide an image for the synopsis. This image should provide a rapid overview of the question addressed in the study but still needs to be kept fairly modest since the image size cannot exceed 550 (width) x 300-600 (height) pixels.

Thank you again for giving us to consider your manuscript for EMBO Reports, I look forward to your minor revision.

Kind regards,

Deniz Senyilmaz Tiebe

--

Deniz Senyilmaz Tiebe, PhD
Senior Scientific Editor
EMBO Reports

Referee #2:

The authors have addressed all the points raised by the referees and I strongly support the publication of this generally very important and excellently revised paper in its present form.

Referee #3:

The authors have addressed all my initial questions and concerns adequately and I recommend publication of the revised version in EMBO reports.

All editorial and formatting issues were resolved by the authors.

Dr. Dejana Mokranjac
LMU Munich
Biozentrum-Cell Biology
Großhadernerstr. 2
Planegg-Martinsried, Bavaria 82152
Germany

Dear Dr. Mokranjac,

Thank you for submitting your revised manuscript. I have now looked at everything and all is fine. Therefore, I am very pleased to accept your manuscript for publication in EMBO Reports.

Congratulations on a nice work!

Kind regards,

Deniz Senyilmaz Tiebe

--

Deniz Senyilmaz Tiebe, PhD
Senior Scientific Editor
EMBO Reports

--
